# A neuronal correlate for time interval estimation in the crow's telencephalon

Melissa Johnston[1,2], Maximilian E. Kirschhock [1,2] & Andreas Nieder [1] ✉

Interval timing, the ability to perceive and estimate durations between events, is essential for many animal behaviors. In mammals, it is linked to specific cortical and sub-cortical brain regions, but its neural basis in birds remains unclear. We trained two male carrion crows on a time estimation task using visual stimuli, cueing them to wait for a minimum duration of 1500 ms, 3000 ms, or 6000 ms before responding to receive a reward. During the task, we recorded activity from single neurons in the nidopallium caudolaterale (NCL), the avian executive telencephalon. Many neurons showed tuning to specific durations, suggesting that time intervals are encoded as abstract magnitudes along an ordered scale. Population-level decoding revealed that NCL activity predicted the crows' intended wait time, independent of the sensory properties of the cues. These findings show that abstract time estimation can emerge from neural architectures different from the mammalian neocortex.

The ability to accurately perceive time is critical for many complex cognitive abilities in both human and non-human animal species. While some cognitive abilities relate to time in a broad sense, such as episodic-like memory (when an event happened) and future planning (allocating time), others, such as delayed decision-making, require the ability to flexibly apply temporal information for goal-directed behavior. Among the latter behaviors are temporal discounting (the tendency to favor immediate rewards over delayed ones, even when waiting would yield a greater benefit)[1], sunk-cost (continued investment in a task based on previously invested resources, e.g., time, despite abandonment being more beneficial)[2], and delayed gratification behaviors (the ability to wait for improved, yet delayed opportunities)[3]. For these behaviors, decision-making often depends on internally monitoring time over a period of seconds or minutes, an ability known as "interval timing"[4].

Evidence from the mammalian brain suggests that both motor and prefrontal regions encode time through a combination of ramping activity and categorical representations. Neurons in motor areas frequently exhibit ramping activity, reflecting either the time elapsed toward an anticipated event or the time remaining until an action is required[5–8]. However, motor regions can also display categorical tuning to specific interval durations, suggesting that they contribute to discrete representations of time[9]. Similarly, neurons in prefrontal regions show sustained or phasic activity linked to abstract temporal representations that underlie higher-order cognitive processes, but these regions have also been found to exhibit ramping activity associated with elapsed time[10–12]. This overlap indicates that both motor and prefrontal areas engage in multiple forms of temporal encoding, contributing to the flexible representation of time during different behavioral tasks.

Birds have evolved a different pallium since they diverged from the mammalian lineage 320 million years ago[13]. The bird pallium is strikingly nuclear and lacks a layered cerebral cortex[13]. Despite this, birds demonstrate sophisticated interval timing in their decision-making behaviors. Hummingbirds, for example, use interval timing to determine when nectar-depleted flowers and feeders are likely to be replenished and are worth revisiting[14,15]. Similarly, scrub jays use time to determine whether previously cached perishable food items have likely degraded past the point of being worth recovering[16]. Interval timing is therefore accessible to birds and their differently evolved telencephalon.

To date, there is limited research examining the neural mechanisms underlying interval timing in birds. A putative candidate region for time estimation is the associative endbrain area termed "nidopallium caudolaterale" (NCL), which is linked to high-level cognition in birds[17–24] and is considered a putative avian analog of the mammalian

[1]Animal Physiology Unit, Institute of Neurobiology, University of Tübingen, Tübingen, Germany. [2]These authors contributed equally: Melissa Johnston, Maximilian E. Kirschhock. ✉e-mail: andreas.nieder@uni-tuebingen.de

prefrontal cortex (PFC)[25,26]. Indeed, some working memory experiments led to the hypothesis that neurons in the NCL are influenced by the passive tracking of time[27,28]. However, the neural foundation of active time estimation in birds remains elusive. In the current experiment, we assess whether the observed behavior aligns with scalar expectancy (i.e., longer estimations resulting in more variation) and then examine the neuronal representations of time intervals in the NCL. The overarching hypothesis is that the uniquely developed avian endbrain, despite lacking a layered pallium, has evolved physiological mechanisms similar to those in mammals to address common computational challenges involved in processing time intervals.

## Results

### Behavior

We trained two carrion crows on a time estimation task whereby two sets of visual stimuli pseudo-randomly cued the crow on three minimum wait durations (1500 ms, 3000 ms, and 6000 ms) required prior to making a response (Fig. 1). Crows were rewarded only if their response occurred after the minimum wait duration had elapsed. Crow 1 and Crow 2 completed 71 and 32 sessions, respectively (Fig. 2). If the crows worked under stimulus control, we would expect a normal distribution of reaction times (RTs) centered around a time point just beyond the minimum time delay. To test this, we fitted Gaussian curves to the RT histograms. For each target duration, RTs were binned into 200 ms intervals. To address the apparent opt-out behavior in the 6000 ms trials, the first two bins (400 ms) were excluded, as we believe these responses did not reflect genuine estimates for the 6000 ms trials. To ensure stable and representative fits, the center of the Gaussian was fixed to the peak of the response time histogram (i.e., the most frequently occurring bin), as this approach reduced the influence of early, low-frequency responses and allowed for more consistent estimation of distribution spread across durations. Fitting resulted in high goodness of fits ($r^2$) averaging 0.99, 0.98, and 0.62 for the 1500 (range for individual sessions: 0.98–0.99), 3000 (range: 0.97–0.99), and 6000 ms (range: 0.36–0.7) wait periods, respectively. The means of the Gaussian curves were 1722, 3201, and 6117 ms, for the 1500 (range: 1795–1930), 3000 (range: 3304–3404), and 6000 ms (range: 6062–6254) wait intervals. As expected for magnitude estimation, the variation in time estimations—measured as the standard deviation (sigma of the Gauss fit)—systematically increased as the target time increased from 299, 642, and 1260 ms for the 1500 (range: 264–381), 3000 (range: 588–780), and 6000 ms (range: 1131–1402) wait periods, respectively (Fig. 2B inset).

### Time preference neurons

We recorded the activity of 409 single units (Crow 1: 265 neurons; Crow 2: 144 neurons) from the NCL (Fig. 3) while crows completed the time estimation task. Based on the inclusion criterion (see "Methods"),

92.36%, 86.72%, and 62.36% of the 1500, 3000, and 6000 ms trials, respectively, were included for further analyses.

To identify absolute timing neurons, we performed a two-way ANOVA on firing rates with stimulus type and target duration as factors (significance threshold of $\alpha = 0.01$), separately for each period of interest for correct trials only. The alignment of the period of interest (1200 ms interval; see "Methods") was either relative to cue offset as a starting point of the analysis window, or relative to response onset as the endpoint of the analysis window. Overall, we found 238 neurons (58.19%; alignment to cue offset: 178 neurons, 43.52%; alignment to response onset: 135 neurons, 33.01%) with a significant main effect of time but no significant main effect of stimulus protocol or significant interaction term. These neurons were labeled "time" neurons. Of these, 75 neurons (31.51% of time neurons) were significant during both the cue offset and response onset aligned periods. For the alignment to cue offset, we found a further 28 neurons (6.85%) that showed a significant effect of cue modality and 17 (4.16%) with a significant cue modality and wait duration interaction. For the alignment to response onset, we found 12 neurons (2.93%) and 5 (1.22%) with a significant cue modality or cue modality and wait duration interaction, respectively. Of the 75 neurons that showed a significant effect of time in both the cue offset and response onset alignment, only 25 changed their preference across the two periods of interest.

The target duration eliciting the highest firing rate in the period of interest per neuron was defined as the neuron's preferred target duration. Figure 3A–C shows three example neurons with a preference for the 1500 (A), 3000 (B), or 6000 ms (C) target duration when the analysis window was aligned to response onset. Of the time neurons whose selectivity was aligned to cue offset, 42 (10.27% of all neurons), 35 (8.56%), and 101 (24.69%) neurons preferred the 1500, 3000, and 6000 ms interval targets, respectively (Fig. 4A). Of the time neurons whose selectivity was aligned to response onset, 69 (16.87%), 24 (5.87%), and 42 (10.27%) neurons preferred the 1500, 3000, and 6000 ms targets, respectively (Fig. 4D). Rather than firing at a steady rate throughout the wait period, time neurons peaked at distinct times during the wait periods, suggesting that ensembles of time neurons form dynamic sequences over time (Fig. 3E, F).

To quantify the difference in firing rates between the preferred and non-preferred target durations, we created tuning curves with normalized activity by setting the maximum activity to the most preferred target time as 100% and activity to the least preferred target time as 0%. The average tuning curves for cue offset and response onset time preference neurons are shown in Fig. 4B, E. For both the cue offset-aligned and response onset-aligned activity, time neurons showed decreased normalized activity for non-preferred wait durations, indicating a clear differentiation between the three wait durations. This is true for both birds (Supplementary Fig. 1). We found a highly significant increase in tuning width with an increase in target

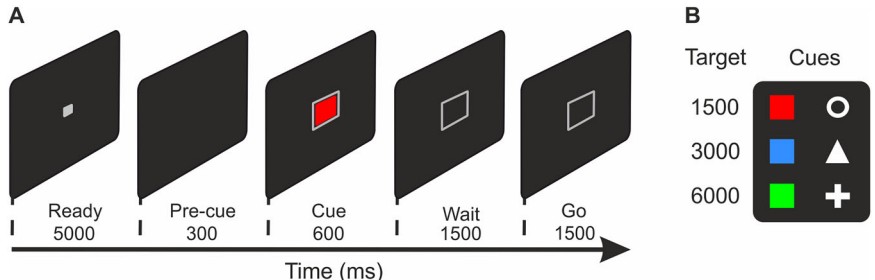

**Fig. 1 | Behavioral protocol. A** Crows were trained on a delayed-response task whereby visual stimuli (**B**) cued the crow to estimate 1500, 3000, or 6000 ms. Each trial began with a "Ready" period (up to 5000 ms) in which the subject moved into position using a light barrier. Moving into position triggered a 300 ms pre-cue period and then 600 ms cue period. Colored squares and shapes served as the cues and instructed the subject on the minimum wait time required before ("Wait" period), leaving the light barrier ("Go" period). Importantly, the "Wait" and "Go" periods matched in duration to account for the increased variance in response time with longer estimates. Here, we present an example of a 1500 ms trial.

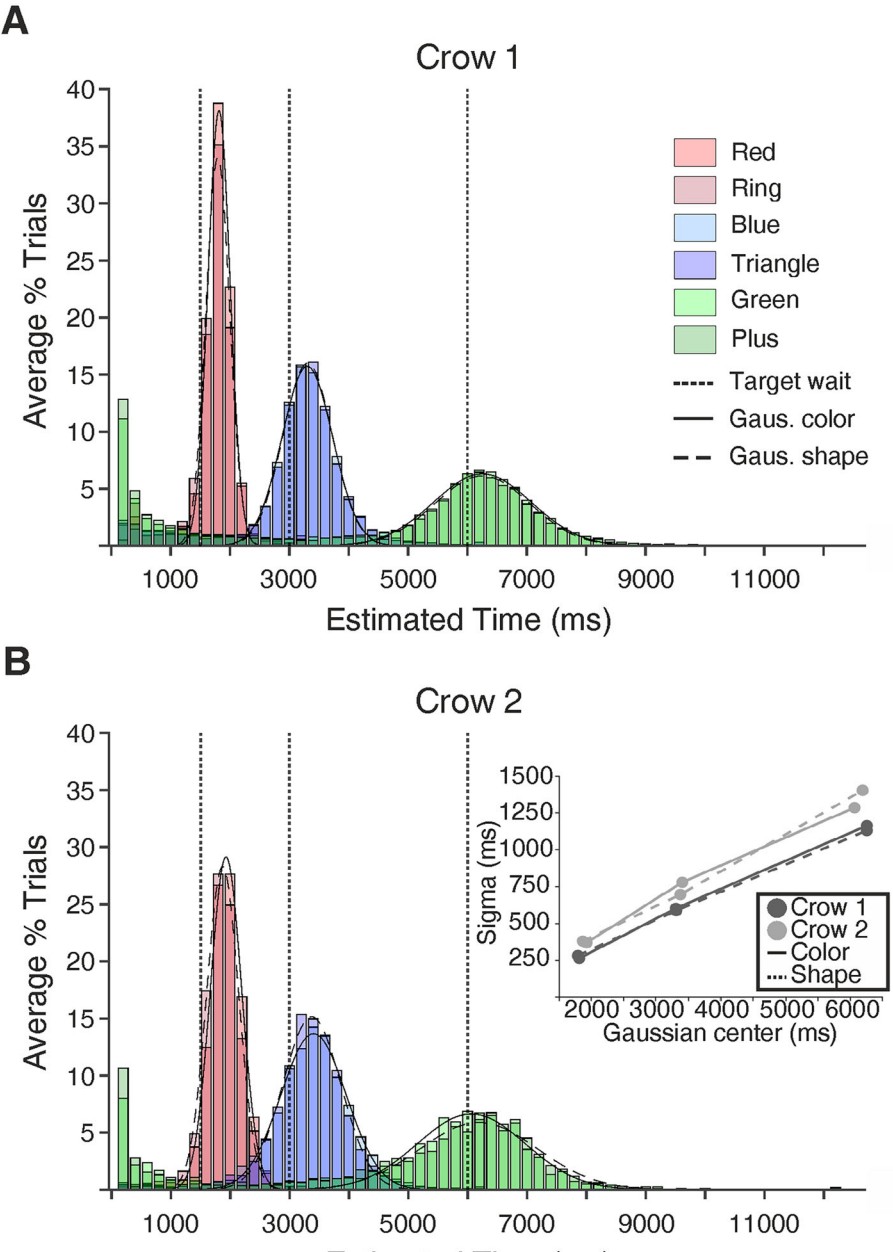

**Fig. 2 | Behavioral results.** Average frequency of reaction times per target estimation (1500, 3000, and 6000 ms) for Crow 1 (**A**; 71 sessions) and Crow 2 (**B**; 32 sessions). The x-axis represents the subject's time estimation, measured as time since cue offset. Variation in time estimations (Sigma of Gauss fit) as a function of the Gaussian centers for Crow 1 and Crow 2 (**B insert**).

duration (Fig. 4C, F; $T_{JT} = 5.54$, $p < 0.001$ for the cue offset alignment; $T_{JT} = 3.87$, $p < 0.001$ for the response onset alignment; Jonckheere-Terpstra trend test). When searching for monotonicity tuning for 1500 ms and 6000 ms neurons, we found only two (4.76%) and one (0.99%) neuron(s) with non-monotonic activity for 1500 ms and 6000 ms cue offset aligned time neurons, respectively. For the response onset time neurons, we found only six (8.7%) and one (2.38%) neuron(s) with non-monotonic activity for 1500 ms and 6000 ms time neurons, respectively. Taken together, these findings indicate that time intervals were represented as related temporal magnitude on an ordered time scale.

**Neuronal correlates of time estimation**
We first searched for indications that single units in the NCL encode the passage of time by systematically increasing firing rates as a

function of time. To that end, we selected neurons in which peak neuronal activity occurred in the 600 ms prior to the response onset and were significantly different from the 600 ms of activity aligned to the cue offset ("increasing" neurons), and fitted linear, exponential, and sigmoidal models. We identified 59 increasing neurons. Of these neurons, only 20 (4.89% of all neurons), 21 (5.89%), and three (0.73%) neurons showed a linear, exponential, or sigmoidal fit significantly better than chance. Importantly, the neurons that showed a significant linear increase also showed a significant exponential increase, meaning the populations overlapped entirely. Therefore, we found very little evidence to suggest single units in the NCL encode the passage of time via ramping activity.

Another way to investigate whether the passage of time is encoded in NCL neurons is to test the temporal scalability of neuronal activity. If activity scales over time, neuronal activity during longer wait

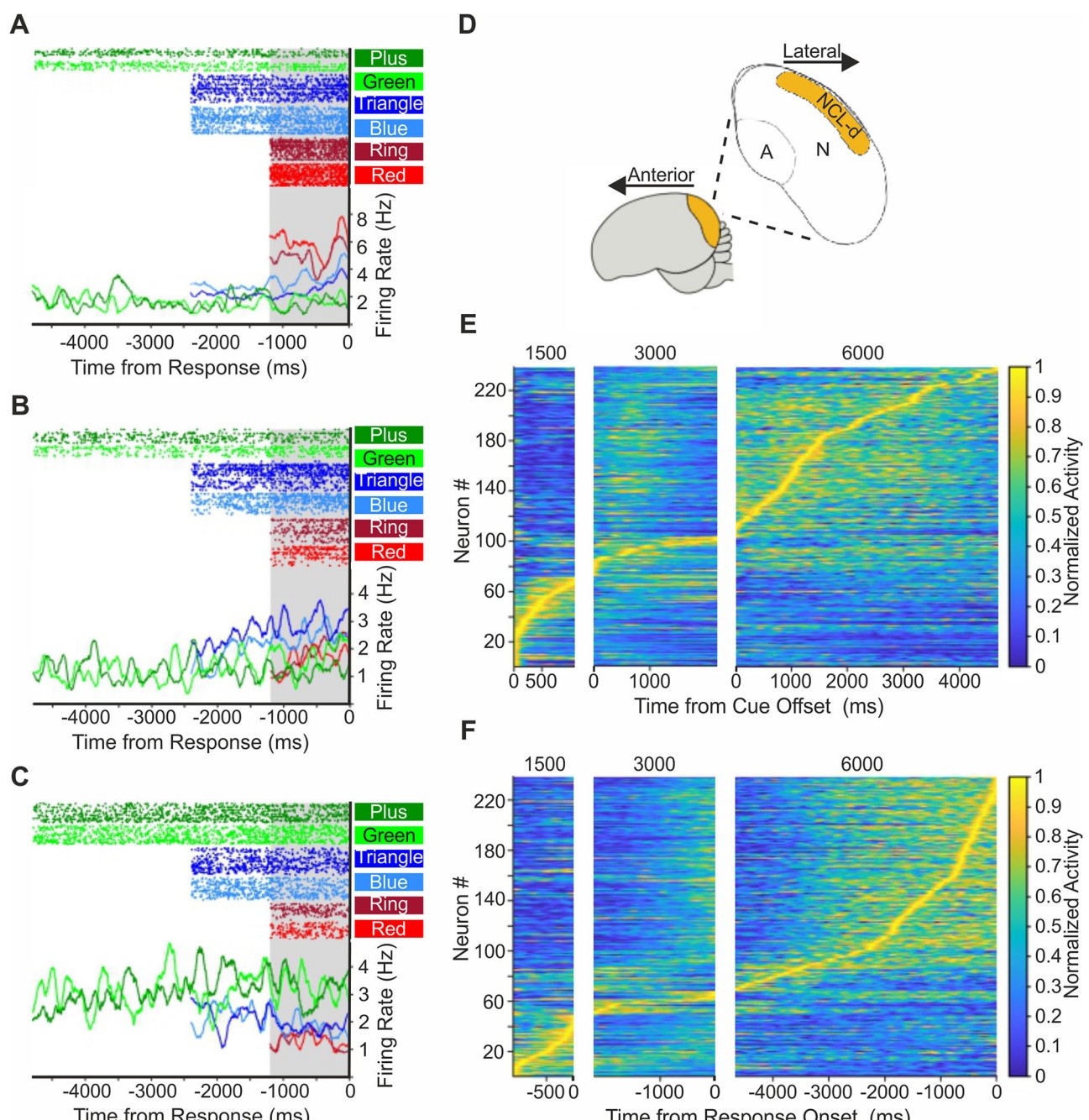

**Fig. 3 | NCL single-unit recordings. A–C** Responses of three example NCL neurons selective to target interval of 1500 ms (**A**), 3000 ms (**B**), and 6000 ms (**C**) during the final 1200 ms of the wait period (alignment to response onset; as indicated by the gray shading). Top: dot-raster histograms with each dot representing one action potential; Bottom: averaged spike density function (activity averaged and smoothed by a 300 ms Gaussian kernel). Time 0 indicates the crow's response onset for each of the three wait intervals. **D** Lateral schematic of a crow brain with a coronal section at the posterior end showing the telencephalic NCLd (yellow). *A*

arcopallium; *N* nidopallium. **E**, **F** Time entries in the wait period during which neurons with a preference for the 1500, 3000, or 6000 ms target duration (each panel) for activity aligned to the cue offset (**E**) or response onset (**F**). Each line represents one neuron showing a significant main effect of time in either cue-offset or response-onset aligned analysis ($n = 220$). Surface color indicates normalized firing rate, with 0 corresponding to the minimum and 1 to the maximum firing rate per neuron across the aligned time window.

periods is expected to be a stretched version of shorter wait periods, or vice versa. Here we searched for temporal scaling between trial types (e.g., from 6000 ms trials to 3000 and 1500 ms trials) using methods reported by Xu and colleagues[29] (see "Methods"). As 3000 ms trials are twice as long as 1500 ms trials, and half as long as 6000 ms trials, the optimal scaling factors for 3000–1500 ms trials and 6000–3000 ms trials should be around 0.5. Similarly, 6000 ms trials

are four times as long as 1500 ms trials and, therefore, should have an optimal scaling factor of around 0.25.

We found that the overall optimal scaling factors for 3000–1500 ms trials and 6000–3000 ms trials were 0.53 and 0.32, respectively (Supplementary Fig. 2). For 6000–1500 ms scaling, the overall optimal scaling factor was 0.23. Despite these seemingly fitting values, these average scaling factors obscured the fact that they

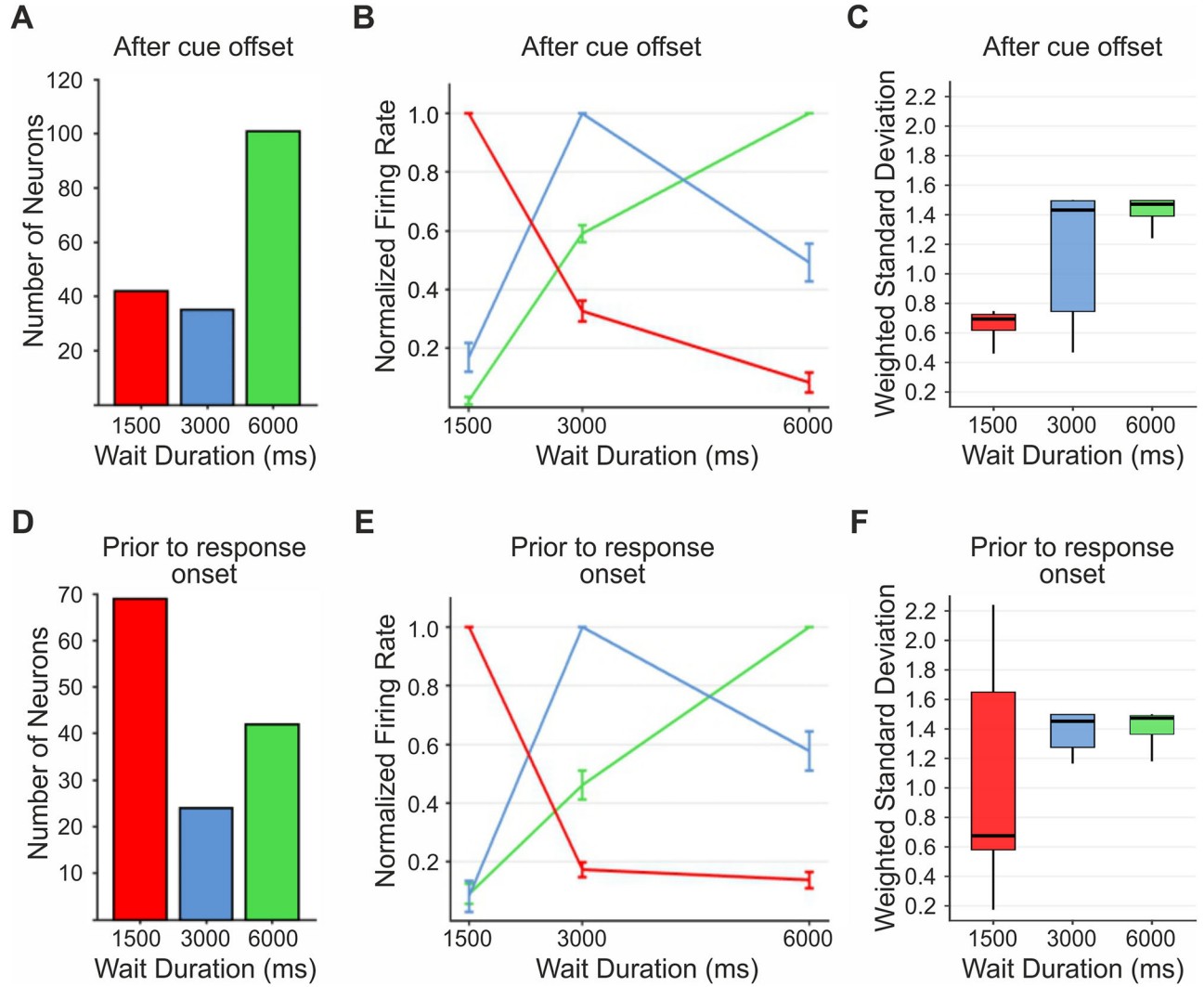

**Fig. 4 | Target wait preference and population tuning curves. A, D** Incidence of neurons preferring either the 1500, 3000, or 6000 ms target duration, a 1200 ms period aligned to either after the cue offset (**A**) or prior to response onset (**D**). **B, E** Population tuning curves for neuronal activity aligned to after the cue offset (**B**, $n = 178$) and prior to response onset (**E**, $n = 135$). Curves were obtained by averaging the normalized tuning curves of all single units selectively preferring either the 1500, 3000, or 6000 ms wait duration from (**A**) and (**D**). Error bars show the standard error of the mean. **C, F** Weighted standard deviation quantifying the width of tuning functions for the population of neurons with a target duration preference when aligned to the cue offset (**C**, $n = 178$) and to the response onset (**F**, $n = 135$). The line within each box represents the median; the box spans the interquartile range (25th to 75th percentile), and the whiskers extend to the minimum and maximum values.

resulted from very diverse scaling factors of individual neurons, giving rise to very broad factor distributions. Indeed, we found that only 27 (8.79% of neurons), 41 (13.36%), and 18 (5.86%) neurons fell within a 10% window of the scaling factors for 3000–1500, 6000–3000, and 6000–1500 ms scales, respectively, but only two neurons fell within the 10% window for all three scaling schemes. Thus, the evaluation of scaling factors suggests that most neurons do not scale their temporal activity profiles.

For a final analysis for determining whether neuronal activity could decode elapsed time, we binned activity of all timing neurons into 20 equal bins (see Methods for a detailed description) for the three wait durations separately and tested whether SVM classifiers trained on 80% of the observations could predict the bin (i.e., elapsed time) of the remaining 20% of observations. We found that the classifier performance was highly accurate in predicting the elapsed time of binned activity (Fig. 5) for all three wait durations for both the cue offset-aligned and response onset-aligned activity. Elapsed time during 6000 ms trials was particularly well estimated, likely due to larger binned periods comprising of larger estimation windows. Therefore,

the subpopulation of timing-selective NCL neurons encoded elapsed time. The same holds true when all neurons irrespective of neuronal selectivity ($n = 409$) were included (Supplementary Fig. 3).

**Target duration decoding**

To test whether the population of NCL neurons contained information regarding the current target duration, we trained and tested linear multiclass SVM classifiers on the firing rates of all recorded neurons, irrespective of time tuning (see "Methods"). We used two different classification approaches. First, a within-protocol approach (training on color cue trials and testing on color cue trials, and the same procedure for shape cue trials) was used to determine whether the neuronal population carried information about the target duration within (color or shape) protocols. Second, in a cross-protocol analysis (training on color cue trials and testing on shape cue trials, and vice versa), we explored whether the neurons contained abstract, cue-independent information about the target wait duration. Specifically, we asked for the cue offset-aligned and the response onset-aligned periods, whether SVM models trained on color stimuli could predict

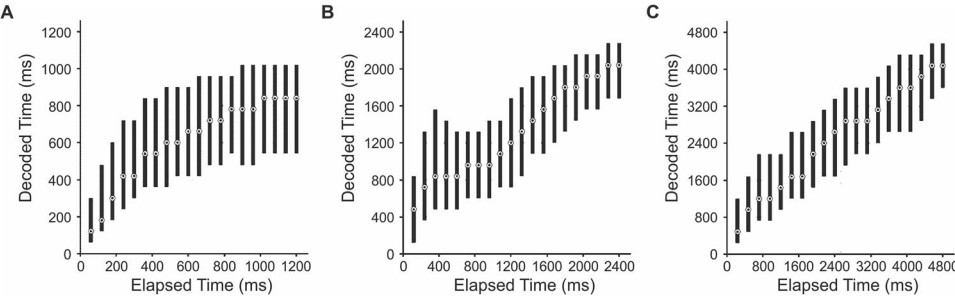

**Fig. 5 | Decoding of elapsed time from timing neurons. A–C** Decoding of elapsed time from the binned activity of timing neurons aligned to cue offset. Neural activity was divided into 20 equal-sized bins for the first 1200 ms of the 1500 ms trials (60 ms; $n$ = 238; **A**), 2400 ms of the 3000 ms trials (120 ms; $n$ = 238; **B**), and 4800 ms of the 6000 ms trials (240 ms; $n$ = 236; **C**). This period represents the earliest response that was considered correct. The dot represents the median, and the bar represents the interquartile range.

the target duration of shape trials and vice versa (wait duration and stimulus protocol).

We started this analysis with firing rates from cue offset to 1200 ms after cue offset. For within-protocol classification, we found that classifier performance was well above chance of 33% (color: 92.33%; shape: 95.36%; Fig. 6A). For across-protocol classification, we observed similarly robust classifier performance (color-to-shape: 89.24%; shape-to-color: 89.33%; Fig. 6A). Similar stable decoding was seen for classifier analyses aligned to the time before the crows' responses, i.e., from 1200 ms before waiting period offset to waiting period offset. Within-protocol classification resulted in highly accurate performance for both cue types (color: 80.76%; shape: 80.86%; Fig. 6C). Cross-protocol classification showed almost equally high decoding performance (color-to-shape: 74.71%; shape-to-color: 77.79%; Fig. 6C). All findings were consistent across both crows (Supplementary Fig. 4). This demonstrates that the neuronal activity of NCL neurons encodes the target duration abstractly, independent of the sensory properties of the cue. This code allowed the classifier to predict the crows' waiting times. With trials now pooled across cue protocol, we trained new SVM classifiers to discriminate target duration based on the activity for the 1200 ms periods aligned separately to the cue offset and response onset, respectively. Once again, the tested SVM model offered the predicted class labels for the test subset of trials as the output. Together with the true class labels for this test subset, a confusion matrix can be constructed (Fig. 6B, D). Values on the diagonal of these confusion matrices, divided by the absolute sum of classifications, yield a percentage measure for decoding accuracy. The resulting confusion matrices in Fig. 6B, D show that the SVM classifiers performed much better than chance (33.33%) for the three target durations (1–3) for both the cue offset (B) and response onset(D) aligned activity; accuracy was at 97% for cue offset aligned activity and at 87.8% for response onset aligned activity. Therefore, neural activity from NCL neurons clearly distinguishes between the three target durations during time estimation.

Next, we explored whether the neuronal activity of time neurons was relevant to the correct time estimation by the crows. To that aim, we trained and tested SVM classifiers on neuronal activity from the first 1200 ms of the wait period of correct trials for time neurons with at least one late error trial for both the 1500 ms and 3000 ms trials. We then tested the model with neuronal activity from previously unseen correct and late error trials (i.e., trials in which the animal failed to respond within the allotted response window despite remaining engaged). We found that the classification of correct trials was nearly perfect (94.4%) for both 1500 ms and 3000 ms trials (Fig. 7A, B). The accuracy of classifiers trained on correct trials was considerably lower when tested on late error trials (54.2%) and below the 99th percentile of shuffled label classifier performance when predicting late-error trials (dashed lines in Fig. 7A, C). This indicates that the population code

during late-error trials differs fundamentally from the population activity observed during correct time estimation, even at the very onset of the crows' timing behavior.

**Temporal dynamics across the population.** We explored the population-level temporal dynamics using three time-resolved neuronal analyses. First, we performed a $\omega^2$ percent explained variance analysis (PEV) analysis (see "Methods" for details). The PEV quantifies the amount of information about different task factors carried by neuronal populations. We found that neurons represented all factors of the instruction stimulus, that is, the target duration, but also the stimulus protocol, and the interaction between both factors (Fig. 8A). Information related to cue properties declined shortly after cue offset, while representation of the target duration remained highly persistent throughout the early wait period. This indicates that the target duration was the most prominently encoded and behaviorally relevant factor.

Next, we trained SVM classifiers on the firing rates from any given time point and tested them during any other time points of new trials. Accuracy was plotted in a confusion matrix spanning the trial times of classifier training against the trial times of classifier testing (Fig. 8B). The classifier was at chance prior to cue onset, but highly accurate throughout the cue and wait periods. This indicates that the wait duration is encoded during both the cue presentation and while the crow is estimating time. Neurons may encode a specific target duration during the wait period and sustain this representation over extended time intervals through persistent firing. This form of coding, referred to as static coding, allows a decoder trained on neuronal activity at one moment in the trial to generalize effectively to other time points. In contrast, dynamic coding involves sparse neuronal firing with rapid changes in tuning over time, meaning that a decoder trained at one moment cannot generalize to subsequent moments in the trial. Our analysis revealed evidence for both coding strategies. Static coding was demonstrated by significant cross-temporal generalization (i.e., square-like pattern), which persisted from the beginning of the cue period through the initial 1800 ms of the wait period. However, a dynamic code based on transiently active neurons along the diagonal pattern is also evident.

Finally, we explored the stability of time selectivity in "time" neurons. To that end, we divided the neuronal activity during the wait periods into 20 equally sized bins for each of the three wait durations. SVM classifiers were then trained on the firing rates from individual bins using a subset of trials and tested on firing rates from previously unseen trials. As with the previous SVM classifier, accuracy was visualized in a confusion matrix, with the trial times of classifier training plotted against the trial times of classifier testing (Fig. 8C). High accuracy along the diagonal reflects a stable encoding of time throughout the wait periods. This finding demonstrates that time selectivity was maintained in a stable and consistent manner across the wait period.

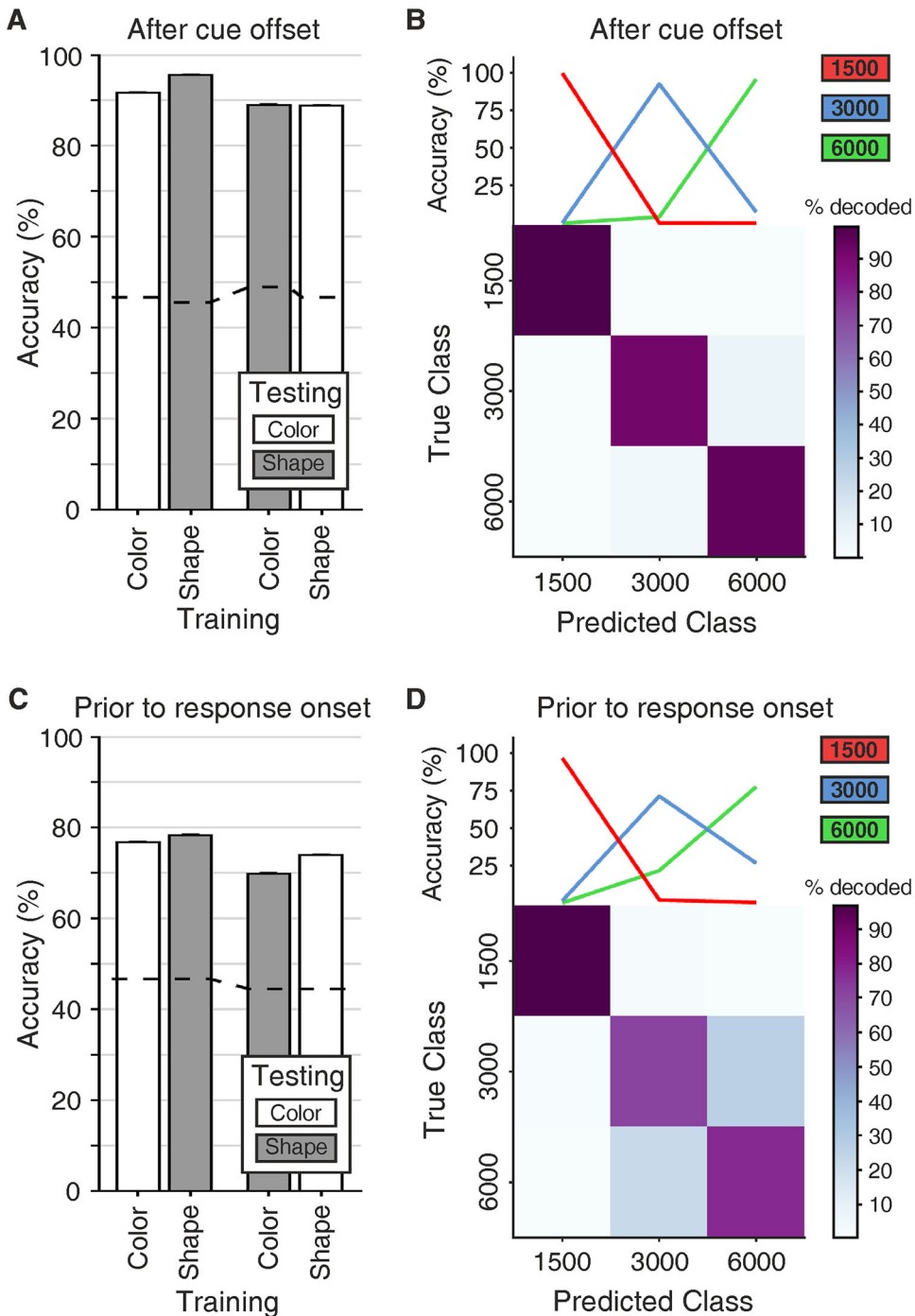

**Fig. 6 | Population decoding of target duration.** Linear multiclass support vector machine (SVM) classifiers using 1200 ms of neuronal activity aligned to either the cue offset (**A**, **B**) or response onset (**C**, **D**). Accuracy of within- (trained on trials from one stimulus protocol and tested on new trials from the same protocol) and across-stimulus protocol performances (trained on trials from one stimulus protocol and tested on new trials from the respective other protocol) classifier prediction performance (**A**, **C**). Dotted lines represent the 99th percentile of shuffled label classifier performance. Error bars indicate the standard error of the mean. Performance of SVM classifiers target duration (**B**, **D**). The top panels show classification performance for each target duration. Bottom panels depict the confusion matrices (averaged over 10-fold cross-validation and 1000 resamples). The scaling of colormaps is the same across both task periods.

To explore the temporal dynamics of how temporal information might be encoded in NCL neurons, we performed a multidimensional state-space analysis. Here, at any moment in time, the activity of $n$ neurons is represented by an $n$-dimensional vector in $n$-dimensional state space. For graphical depiction, the dimensions are reduced to the three dimensions that capture most of the variance of the data. This gives rise to trajectories meandering through state space. While the absolute positions of the trajectories in space are meaningless, the distance between the trajectories signifies coding differences (i.e., discriminability of time intervals) of the population of neurons. The trajectories for each target duration are presented in Fig. 9A. The 1500 ms trajectory diverges noticeably from the 3000 and 6000 ms trajectories immediately following cue onset, whereas divergence between the 3000 and 6000 ms trajectories begins later in the cue period. To quantify the divergence, we calculated the Euclidean distances between each pair of target durations (1500 vs 3000 ms; 1500

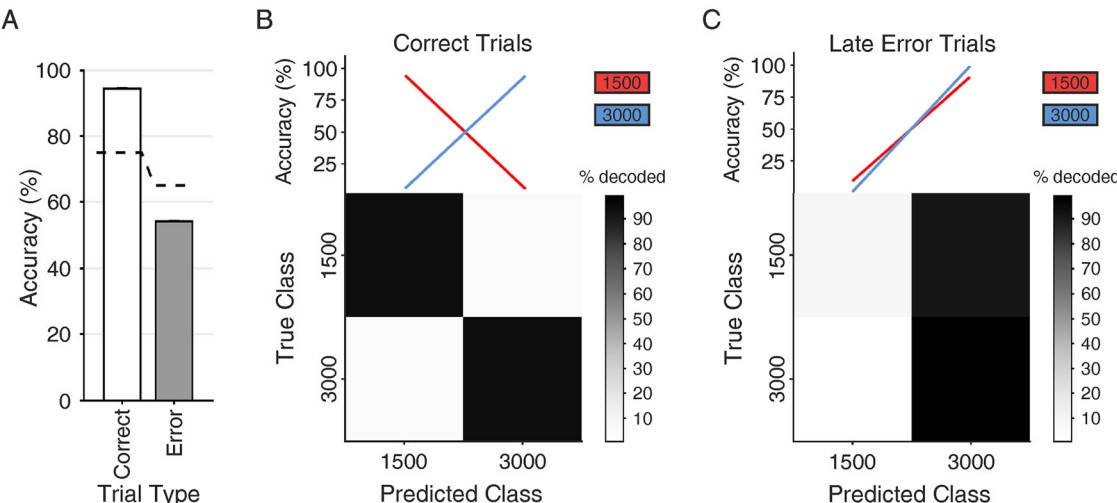

**Fig. 7 | Population decoding of correct and late error trials.** Linear multiclass support vector machine (SVM) classifiers using 1200 ms of neuronal activity from 1500 ms and 3000 ms aligned to the cue offset (**A**–**C**). Accuracy of classifier prediction performance trained on correct trials and tested on new correct or late error trials (**A**, n = 76). Dashed lines represent the 99th percentile of shuffled label classifier performance. Error bars indicate the standard error of the mean. Performance of SVM classifiers for correct (**B**) and incorrect (**C**) trials. Top panels show classification performance, and bottom panels depict the confusion matrices (averaged over 10-fold cross-validation and 1000 resamples).

vs 3000 ms; and 3000 vs 6000 ms) for the trajectories. As shown in Fig. 9B, comparisons to the 1500 ms trajectory result in the largest Euclidean distances (note that the differences in Euclidean distances start after – 600 ms, which is the onset of the cue that the crows associated with different wait times and thus affected state space trajectories). Notably, the difference between 1500 ms and 3000 ms trials is larger than the difference between 3000 ms and 6000 ms trials, even though the absolute difference in time of comparisons is smaller in the former (1500 ms and 3000 ms difference, respectively). The neural trajectories showed a mild increase in amplitude for longer target durations (Fig. 9C)[30]. The trajectory velocity prior to wait onset increased as a function of the lengths of the wait interval (Fig. 9D). This analysis confirmed that NCL neurons categorically signal impending waiting time intervals that can be read out right at the onset of the waiting period from the combined activity of the crows' NCL neurons.

## Discussion
### Crows' behavioral time estimation
In the current experiment, we aimed to investigate the underlying mechanisms of timing behavior in the crow brain using electrophysiological recordings during a time estimation task. Here, crows were trained to make a response after they had perceived 1500, 3000, or 6000 ms had elapsed, as cued by visual stimuli. Overall, we found that the crows' behavioral estimations were centered close to the target durations of 1500, 3000, and 6000 ms. That is, the crows had an accurate concept of the target duration. In addition, we found the estimation distributions were symmetrical, and variation in estimations systematically increased as the target time increased, i.e., estimations were less accurate for longer target times. These two behavioral properties reflect the two key behavioral underpinnings of scalar expectancy theory in animals[31]. However, we also observed that both crows frequently aborted 6000 ms trials within the first second of the wait period. One possible explanation is that the crows adopted a strategic approach to optimize their overall reward rate. By responding early and accepting the 4 s timeout penalty for an incorrect response, the birds may have chosen to bypass the longer trials in favor of shorter ones. Since trial types were randomized, there was a relatively high probability that the subsequent trial would be either a 1500 ms or 3000 ms trial, requiring significantly less waiting time for a reward. This behavior suggests that the crows dynamically adjusted their responses

based on the probabilistic structure of the task, potentially prioritizing efficiency over strict adherence to each trial's demands.

### Sequential encoding of time by NCL neuron ensembles
If neural activity in the NCL were encoding elapsed time, we might expect neurons to resemble those previously reported as motor-timing, relative-timing, or time-accumulator neurons[5,9,12,32]. Motor-timing neurons are categorized as having stable activity that ramps shortly prior to a motor action. Importantly, ramping begins at a similar time point and changes at the same rate in preparation of the action, regardless of the interval duration. Here, activity at the end of the wait durations is the same regardless of the interval duration[32]. Relative-timing neurons are similar to motor-timing neurons, however, the onset and rate of ramping is scaled to the duration of the interval, such that the slope of the ramping activity is steeper for shorter durations[32]. In contrast to motor-timing and relative-timing, the ramping activity of time-accumulator neurons increases at the same rate at the beginning of an interval and continues to rise as a function of the interval duration. That is, longer durations have higher and later ramping peaks[32].

In birds, relative-timing and time-accumulation ramping activity has been reported in NCL neurons from crows[27] and pigeons[28], respectively. In these experiments, however, subjects were not required to actively maintain timing information for successful task completion. Therefore, time was only an implicit factor that may not have been represented by NCL neurons involved in explicit behavior[33].

In the current study, in which timing was explicit, we found very little evidence of systematic ramping activity (motor-timing, relative-timing, or time-accumulation) in the NCL. At the individual level, only a small number of neurons exhibited any form of increasing activity over time, and none showed consistent ramping indicative of elapsed time encoding. Instead, we found time neurons peaked at distinct times during the wait periods, suggesting that ensembles of time neurons form dynamic sequences over time[34–37]. In addition, we examined whether neurons exhibited scaling across wait durations, a potential indicator of elapsed time encoding, but found no evidence of scaling from one duration to another. However, we did find that a classifier was accurate in predicting the passage of time based on patterns of neural activity of the subpopulation of timing neurons, which is in line with findings by Merchant and Averbeck[38], who observed similar temporal

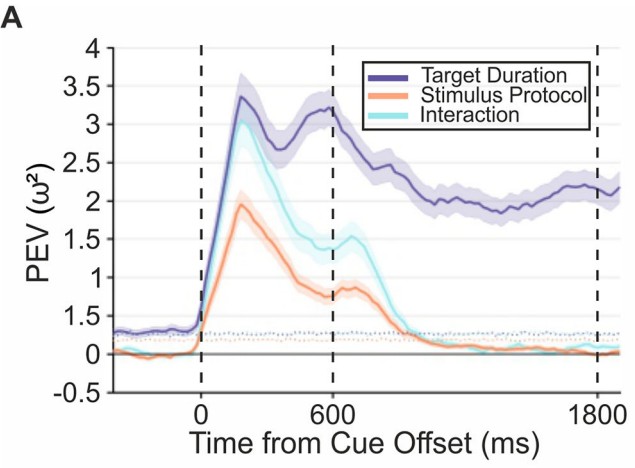

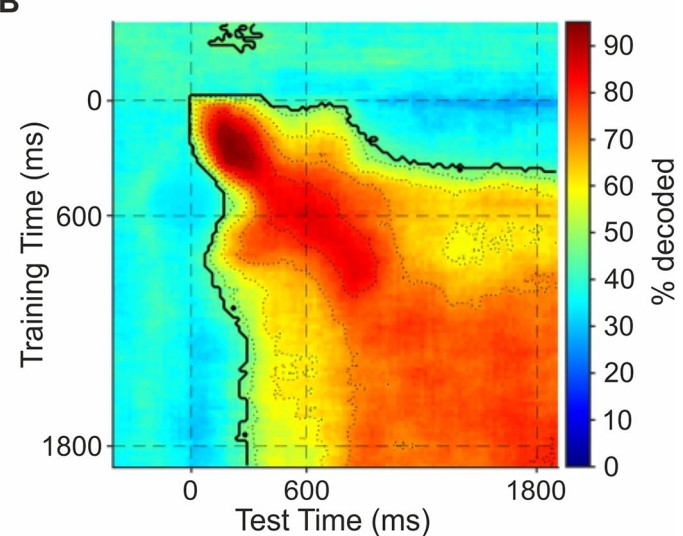

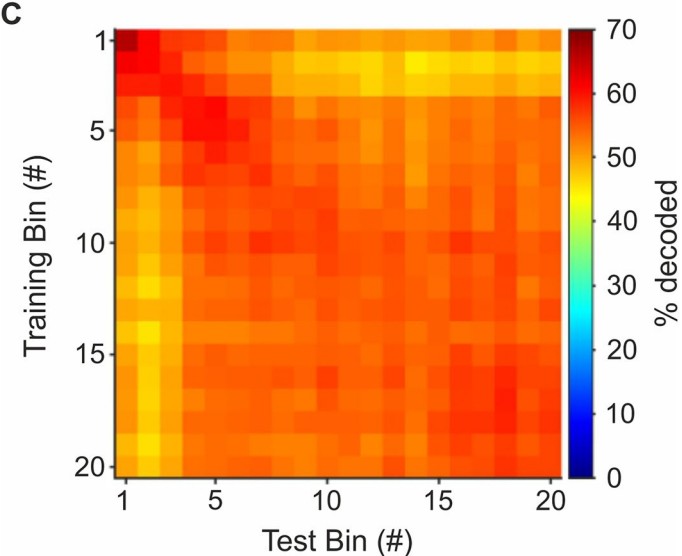

**Fig. 8 | Temporal dynamics and stability.** Information (expressed as time-resolved percent explained variance; solid lines) about factors target duration, stimulus protocol, and their interaction carried by the neuronal population across time (**A**). Dotted lines show the percent explained variance for shuffled trial labels, shaded areas the SEM over resamples. **B** Time-resolved SVM classifier trained and tested with 200 ms bins of neuronal activity for the entire population, starting from pre-cue onset until 1200 ms after wait onset. Straight dashed lines mark the onset of the instruction stimulus period and the start and end of the motor planning period. The area outlined by the thick contour line corresponds to the temporal cluster of time bins significantly above chance level (~ 33%; cluster permutation test, see Methods for details). Dashed contour lines indicate different levels of accuracy (35–65% in steps of 10%). **C** Time-resolved SVM classifier trained and tested using neuronal activity from the wait period of "time" neurons. Activity during the wait period was split into 20 time equally-spaced time-bins (i.e., 20 bins of 60 ms activity for 1500 ms trials, 120 ms bins for 3000 ms trials, and 240 ms bins for 6000 ms trials). For each bin, an SVM model was trained on to distinguish the target duration, then used to predict the target duration of new activity in every other time bin.

expect to see clear overlaps in the state space analysis for this time period if neurons were indeed encoding elapsed time. Conversely, if neural activity had been encoding preparation for a motor response (i.e., motor-timing and relative-timing neurons), we might expect similar findings for the final 1200 ms, i.e., classifiers would have difficulty differentiating activity immediately prior to the decision and activity would overlap in the state-space analysis, as the motor response code should reach a uniform threshold across wait durations.

**Abstract temporal representations in NCL neurons.** In primates, neurons from both motor and prefrontal regions appear to be important for a more general representation of time[9–12]. For example, in time discrimination tasks, where the to-be-compared event durations were unknown to the subject, Marcos and colleagues[39,40] and Genovesio and colleagues[41] show that neurons in prefrontal regions represent time as either "short" or "long" at the time of discrimination (i.e., after the conclusion of both events). In a more complex task involving three durations, neurons exhibited preferences for "short," "medium," or "long" durations at the time of discrimination, with the fewest neurons preferring "medium" durations[10]–a finding we also report here. Similar to what we report in the current study, Yamoto and colleagues[42] found that PFC neurons differentially fired with sustained (rather than ramping) activity for the various estimation periods of time-reproduction task, and Merchant and colleagues[9] report neurons in the medial premotor cortex tuned to specific intervals during single and rhythmic time production tasks. Together, these findings suggest that abstract representations of time are distributed across both higher-order regions and downstream motor areas in the primate brain, rather than being localized to a single cognitive hub for temporal processing. A recent study by Beiran and colleagues[43] investigated how recurrent neural network (RNN) models perform time estimation tasks. The researchers discovered that sustained, tonic inputs can modulate the overall input-output transformation of RNNs by influencing their low-dimensional neural dynamics. This modulation enables the networks to adapt to changing conditions and generalize to novel inputs. The duration-specific, tonic activity observed in NCL neurons may serve a similar function, acting as an internal control signal that supports flexible and adaptive temporal processing. This finding aligns with evidence from monkey studies and corresponds with the computational mechanisms described by Beiran and colleagues[43]. It is not entirely unexpected that the NCL neuron in the current experiment does not encode elapsed time. After all, this brain region is associated with high-level cognition rather than motor functions in birds[17–24], similar to the mammalian PFC[25,26]. Instead, the neuronal data from our study is most in line with the idea that the NCL is important for generating representations of time as an abstract magnitude. We found that NCL neurons showed tuning to preferred

encoding in the monkey medial premotor cortex during a synchronization-continuation task.

At the level of the entire population, the classifiers had no difficulty differentiating between the initial 1200 ms of data from the three wait durations, which would be expected for time-accumulation neurons. Instead, neural data right after cue offset was highly predictive of the current target duration. Due to the same rationale, we would

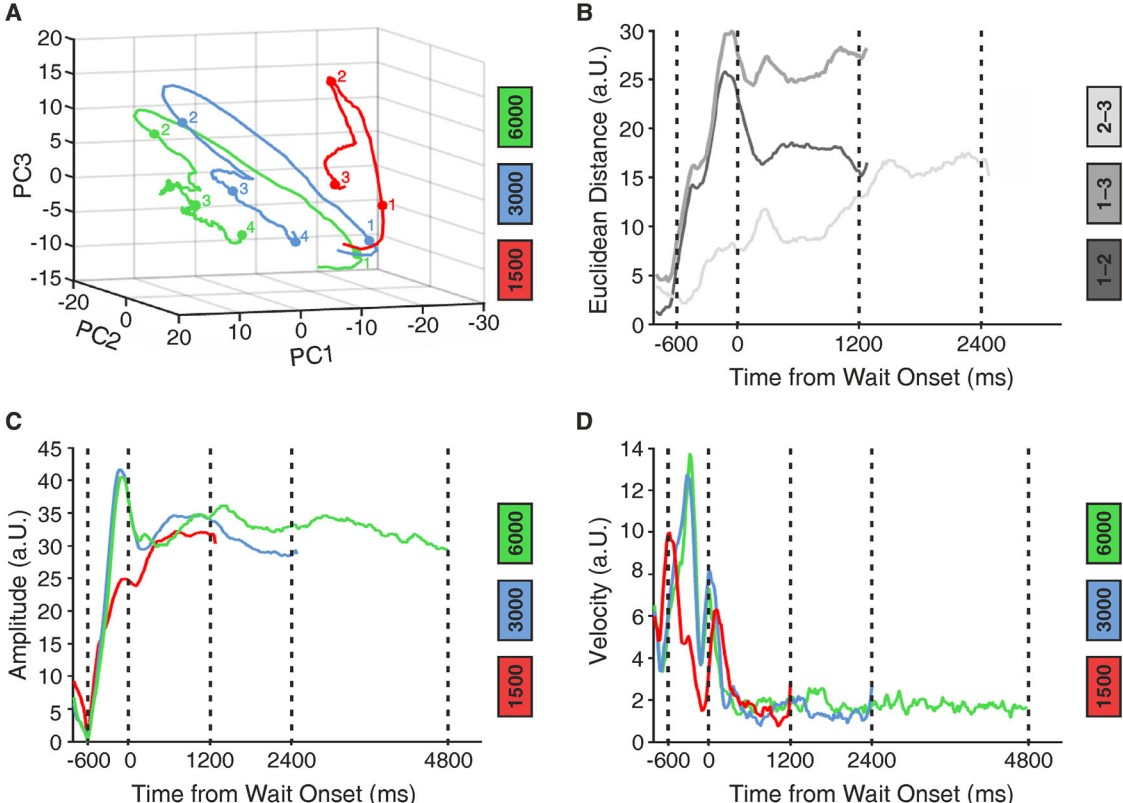

**Fig. 9 | Changes in neuronal activity over time.** Principal component analysis was used to reduce neuronal activity to three dimensions prior to multidimensional state-space analyses, resulting in trajectories for different neuronal states, i.e., the 1500, 3000, and 6000 ms target durations (**A**). The trajectories reflect the instantaneous firing rates of the respective neuronal population as they evolve over time. Numbered points indicate periods of the task: cue onset (1), cue offset (2), 1200 ms into the wait period (3), and 2400 ms into the wait period (4).

**B** Divergence in trajectories was quantified as the Euclidean distance between each pair of target durations (1500 vs 3000 ms; 1500 vs 3000 ms; and 3000 vs 6000 ms) for the trajectories. Time − 600 ms is the cue onset; time 0 ms is the end of the cue and the onset of the wait period. **C** Trajectory amplitudes as a function of time estimation. **D** Trajectory velocities as a function of time estimation. The *y*-axis units in panels B and D are arbitrary units (a.U.).

wait duration, responded abstractly and irrespective of cue modality, and were predictive of the current wait duration. This holds true regardless of whether the activity was aligned to the cue offset (initial 1200 ms) or to the response (final 1200 ms) and was shown to be stable across the entirety of the wait duration. Moreover, we show that information related to cue properties declined shortly after cue offset, while representation of the target duration remained highly persistent. This indicates that the target duration was the most prominently encoded and behaviorally relevant factor. Furthermore, population activity during error trials differed from that observed during correct time estimations, even at the onset of the crows' timing behavior. This suggests that neuronal activity in the NCL was not only predictive but also functionally significant in guiding behavior. Such findings are consistent with previous crow research using other abstract magnitudes, such as numerosity[44,45] and line length[46]. We speculate that neurons encoding elapsed time would be found downstream from the NCL, perhaps in premotor arcopallium.

Overall, our findings demonstrate that crows keep track of time in the range of seconds, likely using a magnitude estimation system[33]. Moreover, we show that NCL neurons represent an accurate internal representation of target durations during explicit timing behavior. However, to establish a definitive causal link between NCL activity and timing, additional studies involving lesion or optogenetic experiments will be necessary. Further understanding such cognitive control over timing behavior and the neural underpinnings will allow us to elucidate how a species with a pallial organization distinct from the mammalian neocortex nonetheless solves shared computational challenges in time processing.

## Methods

### Subjects

Two hand-raised male carrion crows (*Corvus corone corone;* Crow 1 and Crow 2, aged 3 and 7 years, respectively) served as subjects in the current experiment. The crows had previous experience on various match-to-sample and probability judgment tasks. Crows were socially housed in indoor aviaries (see ref. 47 for details). During the experiment, the crows were kept on a controlled feeding protocol and earned food as a reward during training and recording sessions; if necessary, food was supplemented after the daily sessions. Water was provided ad libitum. All procedures were carried out in accordance with European law, the Guidelines for the Care and Use of Laboratory Animals from the National Institutes of Health, and were approved by the responsible local and national authorities (Regierungspräsidium Tübingen).

### Apparatus

Training and recording occurred in a darkened operant conditioning chamber. Leather jesses were used to loosely strap the crow to a wooden perch placed in front of a 15" touchscreen monitor (3 M Microtouch; 60 Hz refresh rate). The chamber was equipped with two cameras (Body: FLIR CM3-U3-13y3M, Lens: Fujinon DF6HA-1B) to monitor the crow's head position (sampling rate 60 Hz). One camera was fixed to the roof, the other one to the left wall of the chamber. Each camera was accompanied by an infrared emitter (Kingbright BL0106-15-28, 940 nm) to track a reflector attached to the crow's head and ensure the crow maintained a central head position in front of the monitor. The headtracking was done using MATLAB (MATLAB

R2017b). Birdseed pellets (Beo Special, Vitakraft) and mealworms (*Tenebrio melitor* larvae) were delivered as reward via a custom-built automated feeder located below the monitor. Loudspeakers (Visation WB10) for auditory feedback were also installed in the chamber. Presentation of stimuli and collection of behavioral responses was managed by the CORTEX system (National Institute for Mental Health, Bethesda, Maryland). Electrophysiological data was recorded in synchrony with stimulus presentation and behavioral responses using a PLEXON MAP system (Plexon Inc., Dallas, Texas).

## Behavioral protocol

Crows were trained on a delayed-response task whereby visual stimuli cued the crow on the minimum wait duration required prior to making a response (Fig. 1). Each trial was initiated when the crow moved its head into the light barrier following the appearance of a ready stimulus (white square) in the middle of the screen. Moving out of this position before the end of the cue period terminated the trial; such trials were excluded from analyses. The ready period was followed by a 300 ms pre-cue period without visual stimulation. Next, one of six stimuli (target cues) were presented in the center of the screen for 600 ms. We used two stimulus protocols (colored squares and white shapes) for each target duration (1500, 3000, and 6000 ms). Red and ring cues indicated the crow must wait at least 1500 ms before responding, while the blue and triangle cues indicated a minimum wait of 3000 ms, and green and plus cues indicated a minimum wait of 6000 ms. In addition, each cue was surrounded by an identical gray square border. During the subsequent wait and go periods, the cue was removed, however, the gray border remained on screen to signal to the crow that the trial was ongoing. After the appropriate amount of time associated with the cue had elapsed, the crow had the same amount of time to make a response by leaving the light barrier (go period), e.g., once the crow had waited 1500 ms after the presentation of the red or ring cue, they had a further 1500 ms to leave the light barrier. Response windows were scaled to accommodate greater variability in responses for larger target estimations, aligning with the principles of scalar expectancy.

Trials in which the crow successfully waited until the go period before leaving the light barrier initiated a positive feedback sound (a ringing 300 ms-sound at approx. 70 dB SPL), as well as a light on the automated feeder that simultaneously delivered a reward. Birdseed pellets (Beo Special, Vitakraft) and mealworms (*Tenebrio melitor* larvae) were used as rewards. If the crow left the light barrier during the wait period, a negative feedback tone sounded (a squeaking 300 ms-sound at approx. 70 dB SPL), a flash of the screen occurred, and a 4000 ms time-out period was initiated. Such trials were not rewarded. A trial was also terminated with negative feedback (sound and light) and a time-out with no reward if no response was made during the go period. All trials were separated by a 500 ms inter-trial interval (ITI) in which nothing was displayed on the black screen. Trial types (red, ring, blue, triangle, green, plus) were randomly intermixed in blocks of 600 correct trials (100 per stimulus), with a delayed retry protocol, such that incorrect trials (too early or too late in responding) were repeated later. Each session consisted of approximately 420–480 correct trials and lasted approximately 2.5 h. The average number of trials experienced (correct and incorrect) in a session by Crow 1 was 129.83 red trials (SD = 10.84), 136.58 ring trials (SD = 17.81), 156.59 blue trials (SD = 74.99), 147.85 triangle trials (SD = 24.66), 151.65 green trials (SD = 29.44), and 151.17 plus trials (SD = 29.53). For Crow 2, the average was 128.57 red trials (SD = 7.70), 140.20 ring trials (SD = 15.38), 184.52 blue trials (SD = 110.70), 162.82 triangle trials (SD = 18.75), 166.92 green trials (SD = 28.66), and 167.01 plus trials (SD = 27.71). Crow 1 and 2 completed one session daily for 71 and 31 days, respectively.

## Training protocol

Crows were initially trained on a delayed match-to-sample (MTS) go/no-go task using the colored cues and a 600 ms delay period. As the crows' performance on the MTS task improved, the delay duration increased until the point at which it matched the final wait duration associated with the cue. Once the crows' performance stabilized with the 1500, 3000, and 6000 ms delay periods, non-match stimuli were removed (i.e., only match trials occurred). Next, the match stimuli were gradually faded until there was no visible comparison stimulus; therefore, at the end of this training phase, the crows were completing the time estimation task with the colored stimuli. Finally, the second set of stimuli (shapes) were added. The new stimuli were first introduced in color with the same instruction, e.g., the ring stimulus was presented in red. Once the crows were accustomed to the shapes of new stimuli, the color was faded out until the shape was completely white.

## Surgery and neuronal recordings

All surgeries were conducted with the animals under general anesthesia once the crows had learnt the task. The crows were anesthetized using a ketamine/xylazine mixture as outlined by Ditz and Nieder[45]. Following the procedure, analgesics were administered to the crows[45]. The head was positioned in a stereotaxic holder customized for crows, with the anterior fixation point (i.e., beak-bar position) set at 45° below the horizontal axis of the instrument. Using stereotaxic coordinates (center of craniotomy: anterior–posterior + 5 mm relative to the interaural line as zero; medial–lateral 13 mm relative to the midline), we chronically implanted two microdrives per implant, each containing four independent electrodes. Crow 1 received one implant on the left hemisphere. Crow 2 received one implant on the right side and a second implant on the opposite hemisphere at a later time. Thus, a maximum of eight electrodes were implanted per hemisphere, depending on the implant configuration. Crows were given three days to rest following surgery before recording began. Crows were not sacrificed after the study. However, the implantation coordinates were verified to lie within the NCL based on histology in previous work[21].

We used glass-coated tungsten microelectrodes with an impedance of 2 MΩ (Alpha Omega). Each recording session began by adjusting the electrodes until a clear neuronal signal (at least 3:1 signal-to-noise ratio) was identified on at least one channel. Neurons were not preselected for task involvement. Each microdrive had a range of ~ 6 mm, allowing recordings from the NCL at varying depths over several weeks. Each microdrive moved all four electrodes simultaneously. On average, in Crow 1, the two microdrives were advanced ~ 93.6 μm/day and ~ 67.4 μm/day, while in Crow 2, the microdrives were advanced ~ 91.7 μm/day and ~ 105.6 μm/day.

During each session, the birds were placed in the recording setup, and a head stage with an amplifier was attached to the implanted connector on the bird's head. This was connected to a second amplifier/filter and the Plexon MAP box outside the setup via a cable positioned above and behind the bird's head (all components from Plexon). Signal amplification, filtering, and digitization of spike waveforms were handled by the Plexon system. Spectral filtering was achieved through a combined preamplifier filter (150 Hz–8 kHz, 1-pole low-cut, 3-pole high-cut) and a main filter (250 Hz, 2-pole low-cut filter). Amplification levels were individually set per channel, generally at around 20,000x gain. Spike waveforms were sampled at 40 kHz (one entry every 25 μs) over a duration of 800 μs, yielding a 32-element vector. Spikes were manually sorted into single-unit waveforms using Plexon's Offline Sorter, based on two-dimensional plots of waveform features such as peak amplitude, trough, and the first two principal components (PC1 and PC2). We only considered those clusters as single units which were clearly separable (at least 3:1 signal-to-noise ratio) from the noise throughout the sorting time window and had less than 0.1% refractory period violations (assuming a 1 ms refractory period) in their inter-spike intervals.

## Neuronal analyses

All neuronal analyses were conducted in MATLAB (Version R2022b). For analyses, we only included neurons that were recorded for at least eight correct trials for each cue and had an average firing rate of at least 0.5 Hz during the combined pre-cue, cue, and first 1200 ms of the wait period. Regardless of whether the crow was rewarded (for waiting beyond the minimum time delay) or not, it is likely that at the time of response, the subject perceived that the appropriate amount of time had passed, as suggested by the response time distributions in Fig. 2. Therefore, for neuronal analyses, we classified trials as "correct" if they occurred after a time 20% earlier than the target time in a trial, i.e., from 1200 ms for target time 1500 ms trials, from 2400 ms for target time 3000 ms, and from 4800 ms for 6000 ms target time trials. Trials in which the crows waited at least 1200 ms but less than 2400 ms for cued 3000 ms trials, and 4800 ms for cued 6000 ms trials were classified as "early error" trials. Additionally, trials in which the crows did not answer within the response period (i.e., within 1500 ms, 3000 ms, or 6000 ms) were classified as "late error trials". Unless stated otherwise, analyses were conducted using "correct" trials only.

## Single-unit analyses

When searching for categorical representations of time at the single-unit level, we examined neuronal data from 1200 ms (minimum time for correct response for 1500 ms trials) windows aligned to two periods of interest—first, after the cue offset (0–1200 ms of wait period) and second, prior to crow's response onset (1200–0 ms prior to the response)—to account for the varying length of estimation times. For each period of interest, we performed a two-way ANOVA with stimulus protocol (color and shape) and target time (1500, 3000, and 6000 ms) as factors, with a significance threshold of $\alpha = 0.01$. A significant main effect of time, but no significant main effect of stimulus protocol or significant interaction term, were labeled "time" neurons. To quantify the difference in firing rates between the preferred and non-preferred target durations, we created tuning curves with normalized activity by setting the maximum activity to the most preferred target time as 100% and activity to the least preferred target time as 0%. We calculated the weighted standard deviation of each tuning curve as a measure of tuning width. For each curve, we calculated the weighted standard deviation of its x-values, where the corresponding normalized response values (y-values) served as weights. We used a Jonckheere-Terpstra trend test ($\alpha = 0.05$) to determine whether tuning curve widths increased across the increasing target durations and to search for monotonicity of the tuning curves of target durations 1500 ms and 6000 ms.

To determine whether the time neurons form neural sequences (i.e., show a progressive pattern of activation filling time intervals), we generated surface plots for cue offset and response onset aligned activity. Here, the activity of each time neuron was smoothed (200 ms Gauss window) and normalized to its peak activity across all three wait periods (1200 ms, 2400 ms, 4800 ms, respectively).

We also explored the possibility that neuronal activity encapsulated a correlate of elapsed time. First, we searched for systematic changes in neuronal activity during the wait period. Here, we took each neuron that reached the aforementioned neuronal analyses inclusion criteria ($n = 409$) and compared the initial 600 ms of cue offset-aligned activity to the final 600 ms of activity aligned to the response onset (i.e., initial and final 600 ms of wait period) using a paired $t$test ($\alpha = 0.05$). If peak activity occurred in the 600 ms prior to the response onset and was significantly different from the 600 ms of activity aligned to the cue offset, neuronal activity increased and the neuron was classified as a "increasing". To determine whether activity changed systematically across the wait period, we first normalized activity binned in 100 ms time bins between the minimum and maximum firing rate and then fit each neuron with linear, exponential, and sigmoidal models. We evaluated the goodness of fit ($r^2$) compared to the

distribution of $r^2$ values from 1000 shuffled label fits for each cell and each of the fitted functions (i.e., 59,000 shuffled values from 59 neurons for each of the three functions).

Second, we set out to determine whether neuronal activity from longer wait durations scales down to shorter wait durations using each neuron that reached the aforementioned neuronal analyses inclusion criteria ($n = 409$). Here we compressed the first 4800 ms of activity (i.e., cue offset aligned) from 6000 ms trials to either 2400 and 1200 ms (i.e., to 1500 and 3000 ms trial activity). In addition, we scaled the first 2400 ms of activity from 3000 ms trials to 1200 ms. Data was organized into 10 ms bins and scaling was done using various scaling factors (0.3–3 in steps of 0.25) and calculated the associated mean squared error. The optimal scaling factor was that with the smallest difference (mean squared error) between the scaled and scaled-to activity, with larger scaling factors indicating less scaling.

## Population analyses

For all population analyses, we pooled data across sessions. On the subpopulation of timing-selective neurons, we trained and tested linear multiclass support vector machine (SVM) classifiers using neural activity of timing neurons during correct trials to determine whether neural activity could accurately predict elapsed time. We conducted this analysis separately for each target duration (1500 ms, 3000 ms, and 6000 ms), and included only timing-selective neurons with at least 20 correct trials for that duration ($n = 238$, 238, and 236 neurons, respectively). For each duration, we extracted the first 1200 ms, 2400 ms, and 4800 ms of cue offset-aligned neural activity of all timing-selective neurons, respectively, and divided it into 20 equal-duration time bins (i.e., 60 ms, 120 ms, and 240 ms per bin for 1500 ms, 3000 ms, and 6000 ms trials, respectively). Each combination of trial and bin was considered an observation, resulting in 400 (20 trials x 20 time bins) observations for each neuron. Next, we calculated the average normalized firing rate per observation, which served as the feature for classification. We then z-scored each neuron's firing rates using the mean and standard deviation of the training set. Each SVM classifier was trained to predict the elapsed time bin (i.e., one of 20 time classes) from the neurons' binned activities. The category (class) label for each observation was the bin number (1–20), representing elapsed time. Each model was trained on 80% of the observations (i.e., 320 observations) to predict the bin (i.e., elapsed time) of the remaining 20% of (80) observations. To account for multiclass classification arising from the 20 classes, we used one-vs.-one transformation to binary classification provided by the used models. This analysis allowed us to assess whether elapsed time could be reliably decoded from the subpopulation of timing-selective NCL neurons, without assuming any specific functional form of firing over time. Five-fold cross-validation was performed, resulting in 320 trials for training and 80 trials for testing. The procedure was repeated 1000 times, with a new set of randomly drawn observations and new cross-validation splits each time. The tested SVM models output the predicted class labels for the test subset of observations, i.e., a list of predicted bins (predicted elapsed time) for all 80 observations in the test set. These 80 observations contain activity from each of the 20 time bins exactly four times. From this output, we accumulated separately for all 20 bins the number of times each elapsed time bin was predicted by the models, over all cross-validation runs and repeats. This results in one distribution of predicted elapsed times for each of the 20 bins, containing 20,000 values (1000 repeats * 5 cross-validation runs * 4 observations per bin) each. These distributions are visualized as median and interquartile ranges in Fig. 5.

We trained and tested SVM classifiers on the entire population of recorded neurons ($n = 409$) to test whether the NCL contained information regarding the target duration. First, we used an SVM classifier to first test whether SVM models trained on color stimuli could predict the target duration of shape trials and vice versa to determine whether

neuronal activity represents an abstract and instruction-specific coding of the target duration irrespective of the sensory properties of the cue. For this SVM classifier analysis, the category (class) label was the target duration of a trial, and the average firing rate in the specified time window and trial category (correct or incorrect) was the feature used for classification. We only included neurons with at least 15 trials per class (target durations) in this analysis. The classifier models used one-versus-one classification to deal with three classes (1500, 3000, and 6000 ms target durations). Five-fold cross-validation was performed, resulting in 12 trials for training and three trials for testing per class. Trial firing rates were z-scored within each cross-validation repetition. Only the training trials were used for deriving z-scores. The procedure was repeated 1000 times, with a new set of randomly drawn trials for each cell and new cross-validation splits each time. We tested the chance-level performance of the classifier by repeating the procedure with shuffled label assignments. The tested SVM model offers the predicted class labels for the test subset of trials as the output. We trained a second SVM classifier to discriminate target duration based on the activity for the 1200 ms periods aligned separately to the cue offset and response onset. Class labels, SVM training and testing procedures were the same as the initial classifier, except for the inclusion criteria and cross-validation. Here, we used all neurons with at least 30 trials per target duration, which allowed us to use a ten-fold cross-validation, resulting in 27 trials for training and three trials for testing per class.

To assess the behavioral relevance of time neurons, we compared their activity between correct trials and late error trials. We focused specifically on late errors, as these reliably reflect failures in timing behavior, whereas early errors may result from intentional trial abortion or ambiguous causes, making their interpretation less consistent. Late errors were defined as trials in which the crow failed to respond within the allowed response window, despite remaining behaviorally engaged in the task. Engagement was confirmed by the fact that crows had to maintain a specific posture in front of the screen throughout the wait and response period, ruling out non-engagement as a cause. Due to the relatively low number of late errors—and the near absence of such errors for the third wait duration (as this would require waiting over 12 s)—we limited the analysis to neurons with at least one late error trial for both the 1500 ms and 3000 ms cued wait intervals ($n = 76$). An SVM classifier was trained on cue offset-aligned activity (0–1200 ms after cue offset) from correct trials. The trained model was then tested to determine the class (i.e., 1500 ms or 3000 ms trial) of previously unseen activity, originating from correct trials or late error trials. The analysis was performed using 10-fold cross-validation and repeated 1000 times, with correct and late error trials resampled for each iteration. All other procedures followed those used in the earlier classifier analyses.

To quantify the amount of information about target duration carried by neurons throughout the trial, we conducted a percent explained variance (PEV) analysis. The PEV measures the extent to which the variance in neuronal firing rates can be explained by task-related factors (i.e., target duration, stimulus protocol), regardless of selectivity. For this analysis, we included all recorded neurons that had a minimum of 15 trials per wait duration for each of the two stimulus protocols ($n = 374$). We applied two-way (target duration and stimulus protocol) sliding-window ANOVAs, using a 200 ms window that advanced in 20 ms steps, starting from pre-cue onset and extending to 1800 ms after cue offset. The variance attributed to wait duration, stimulus protocol, and their interaction over time was calculated as $\omega^2$ using the formula:

$$\omega^2 = \frac{SS_{term} - df \times MS_{error}}{SS_{total} + MS_{error}} \times 100 \tag{1}$$

where $SS_{term}$ is the sum-of-squares for the factor of interest (target duration, stimulus protocol), $SS_{total}$ is the total sum-of-squares, $df$ is the

degrees of freedom, and $MS_{error}$ is the mean squared error. To extract the population-level PEV over time for the term of interest, we averaged the $\omega^2$ values across all neurons. This procedure was repeated 20 times, each time using a new set of randomly drawn trials. The mean PEV and standard error of the mean (SEM) were then calculated across the resamples. To establish a baseline PEV, we repeated the same analysis with shuffled trial labels in the ANOVAs. For each resample, trial labels were shuffled 50 times, resulting in a total of 1000 reshuffles (20 resamples × 50 shuffles). The baseline PEV was derived from this shuffled data to compare against the actual PEV values.

To investigate dynamic coding at the population level, we performed a cross-temporal SVM classifier analysis. For this analysis, we included all neurons with at least 20 correct trials per target duration ($n = 404$). We applied a sliding-window approach (200 ms window length, advancing in 20 ms steps), starting from pre-cue onset and continuing until 1800 ms after cue offset. In each time window, we trained a linear multi-class SVM model. Using a 10-fold cross-validation framework, the data were divided into 10 equal parts. Specifically, firing rates from 18 trials per class within the respective time window were used to train the model, while the remaining 2 trials per class were used to test the model against firing rates from all other time windows. This process generated a two-dimensional accuracy matrix, where the first dimension corresponds to the time bins used for training the classifier and the second dimension to the time bins used for testing. The training and testing procedures were repeated 10 times, with a different split of trials for each iteration. As with previous SVM classifiers, and firing rates were z-scored prior to training and testing. The entire procedure was repeated 20 times, each time using a new subset of randomly sampled trials. As with previous SVM classifiers, the category (class) label was the target duration of a trial, and the average firing rate in the specified time window was the feature used for classification. To evaluate chance-level accuracy for the cross-temporal classifier, we conducted a cluster permutation test. This involved repeating the entire procedure (including 10-fold cross-validation and z-scoring) using permuted trial labels, with 50 shuffles per resample (resulting in 50 shuffles × 20 resamples = 1000 reshuffles to estimate chance-level accuracy). We then compared the mean of the true accuracy values (averaged over resamples) to the distribution of randomized values at each time bin (corresponding to each pixel in the 2D accuracy matrix; $\alpha_{cluster} = 1\%$). This process was repeated for all 1000 permuted accuracy matrices. Next, neighboring pixels that exceeded the initial significance threshold were grouped into "candidate clusters." The size of each cluster—determined by the number of adjacent significant pixels—was measured for both the true data and shuffled data. These cluster sizes formed a distribution that allowed us to assess the significance of the true accuracy clusters ($\alpha_{rank} = 1\%$).

We further assessed the stability of time selectivity in the population of time-selective neurons using an SVM classifier. Only time-selective neurons with at least 20 correct trials per target duration were included ($n = 236$). Neuronal activity during the wait period was divided into 20 equally spaced, non-overlapping time bins for each target duration separately: 60 ms bins for the first 1200 ms of 1500 ms trials, 120 ms bins for the first 2400 ms of 3000 ms trials, and 240 ms bins for the first 4800 ms of 6000 ms trials. For each time bin, an SVM model was trained to classify the cued time interval (1500, 3000, or 6000 ms) as the class label, and the trained model was then used to predict the cued interval using neuronal activity as the feature from every other time bin. This procedure employed five-fold cross-validation and was repeated 1000 times, each time with newly drawn trials. To determine chance-level performance, the entire analysis was repeated using shuffled trial labels across 1000 permutations. The resulting accuracy matrix reflects how well a classifier trained on activity from time bin y can predict the cued interval in time bin x, with each pixel representing the corresponding classification accuracy.

To explore the temporal dynamics of how temporal information might be encoded in NCL neurons, we performed a multidimensional state-space analysis on the population of recorded neurons. At each point in time, the activity of the population of neurons is defined by a point in $n$-dimensional space. Dimensionality reduction to the first three dimensions (using a principal component analysis) results in trajectories for different neuronal states, i.e., the target durations. The trajectories reflect the instantaneous firing rates of the respective neuronal population as they evolve over time. For the population state-space analysis, we selected neurons that fired during a window beginning – 900 ms prior to cue onset until 1200, 2400, and 4800 ms into the wait period for the three target durations, resulting in windows of 2100, 4700, and 5700 ms, respectively, and had at least 30 trials per target duration ($n = 380$). For each neuron, spike trains were averaged across trials for each duration, smoothed (200 ms, in 20 ms steps), and neuron-wise z-scored before calculating the principal components to prevent the state-space dynamics representing only a few highly discriminative neurons. In addition, we measured the amplitude and speed of the state-space trajectories. Amplitude was calculated as the Euclidean distance of each point in the neural (3D) trajectory to the point of cue onset of the trajectory. Velocity was determined by calculating the change in position along the trajectory—specifically, the Euclidean distance between consecutive points—divided by the time step between them (in seconds). For visualization, speed was smoothed over time using a 5-bin boxcar window.

### Reporting summary

Further information on research design is available in the Nature Portfolio Reporting Summary linked to this article.

## Data availability

The data that support the findings of this study are available from the corresponding author upon request. The data can only be made available from the authors on request because the data awaits further analysis. Source data are provided with this paper.

## Code availability

The code that supports the findings of this study is available from the corresponding author upon request. Only customarily available code was used.

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

## Acknowledgements

This research was supported by an Alexander v. Humboldt Foundation Postdoctoral Fellowship to M.J., and DFG grant NI 618/11-1 to A.N.

## Author contributions
M.J. designed and performed experiments, analyzed data, and wrote the paper. M.E.K. analyzed data and wrote the paper. A.N. designed experiments, evaluated data, wrote the paper, and supervised the study.

## Funding

## Competing interests
The authors declare no competing interests.
