## [Transparent Peer Review file · Nature Communications]

A neuronal correlate for time interval estimation in the crow's telencephalon

Corresponding Author: Professor Andreas Nieder

Version 0:

Reviewer comments:

Reviewer #1

(Remarks to the Author)

This paper investigates the neural correlates of interval timing in the nidopallium caudolaterale of crows executing a single interval production task using a range of durations between 1500 to 6000ms. The animals were able to reproduce the three target intervals with a precision that followed the scalar property of interval timing. The neurons showed no ramping activity nor clear temporal scaling. In contrast, a large population of cells was tuned to one of the three tested intervals, and accurate target duration classification was observed using their activity. Finally, the neural population trajectories showed distinctive dynamics for each target interval.

This is an interesting and well performed study, providing novel information regarding the neural signals associated with timing an action in the range of seconds in the pallium of crows. The behavior of the crows is very robust, and neurophysiology shows that neurons encode time using an interval tuning signal. The idea of using different visual cues to trigger the same timing performance is very clever. I have four main concerns. First, it is not clear how the authors computed the reaction times, they start to count after the cue offset? If this is the case the right tail of the RT distributions includes incorrect trials. Please make sure this is clear. In addition, it would be nice to know what happened with the responses of tuned neurons during error trials. Second, it would be interesting to determine whether the activity of timing neurons can decode elapsed time from the binned activity of neurons (Merchant and Averbeck, 2017). Third, I recommend that the authors perform analyses to determine whether the timing neurons form neural sequences, where neurons show a progressive pattern of activation that fills the time intervals (Crowe et al., J Neurosci; Gouvea et al., 2015 eLife; Zhou, et al., 2022. PLOS Computational Biology; Gamez et al., 2019 Plos Biol). Finally, it is important to measure the changes in amplitude, the speed and the final position of the neural trajectories for the 3 target durations. It has been shown that elapsed time is encoded in the final position of the trajectories (Sohn et al., Neuron 2019; Bi and Zhou, 2020 PNAS; Merchant and Perez, Nature machine intelligence 2020), whereas the speed and amplitude can encode prediction of future events (Wang et al., 2018 Nat Neurosci; Betancourt et al., Cell Rep 2023).

Minor comments.

It would be useful that the x-axis of figure 2 show the target interval and depict the means of the gaussian distributions. Please also report the number of neurons with significant main effect of stimulus protocol and the interaction target duration x stimulus protocol.

It would be nice to have figure showing the limited scaling in the neural responses.

Did you find an increase of the width of duration tuning as a function of the interval?

Did the authors find differences between crow individuals in duration tuning, the decoding and neural trajectories?

Why the Euclidean distance in Figure 6B showed a peak before the time estimation onset?

In the discussion, please consider that the paper of Merchant et al., 2011 PNAS describes the properties of ramping neurons in medial premotor areas, while the paper Merchant et al., 2013 J Neuroscience deals with interval tuning during single interval or rhythmic time production tasks.

Typos line 82 ,2019) However, t

Figure 4a: Nuerons

Reviewer #2

(Remarks to the Author)

This study by Johnston, Kirschhock, and Nieder examines how time is represented in the brains of carrion crows performing a well-trained cued delay task. The neural recordings were obtained from the caudolateral nidopallium, which is thought to be the avian equivalent of prefrontal cortex in mammals. The behavioral performance was consistent with a magnitude-based representation of time, in that the variance of the errors scaled with the duration of the interval. While the birds were waiting, a subset of NCL neurons exhibited tuning for specific delay durations, and the population activity jointly encoded the cued delay time with a high degree of accuracy, which suggests that NCL prospectively encodes delay time, possibly as a dimensional quantity. This behavior is consistent with what has been seen in primate prefrontal cortex, and in contrast to the ramping activity seen in downstream motor areas.

A finding that the crow brain represents time similarly to the brains of primates is important because of how radically different the neural architectures are of these widely diverged species. For the most part, the data supporting this claim are convincing. The number of animals and number of units are appropriate given the difficulty of the experiments, and the experimental and analytical methods are generally sound. The manuscript is well-written and clear except for a significant structural issue described below. Most of the weaknesses are minor and should be easy to address in a revision. However, the results as presented fall short on some major points.

Scalar expectancy theory makes two predictions about the neural representation of magnitudes. First, it should be abstract, independent of the physical characteristics of the stimuli that cue the behavior. This is very clearly supported by the data in this study: there are a large number of time cells that encode the duration signified by the cue without depending on whether the duration was cued with color or shape, and a classifier trained on population activity during one type of cue successfully generalized to the other type of cue. Second, as the authors note around line 317, the representation should be ordered and dimensional, with tuning curves that have a clear peak on the time scale and fall off more or less monotonically in either direction. This is the point that I am not sure has been convincingly demonstrated, for three main reasons.

The first is that time cells are included and classified based on measures that are later used as dependent measures to support the claim that the tuning curves have a particular shape. The inclusion criterion, a significant main effect of target time in a two-way ANOVA, is going to select for neurons that have a peaked tuning curve. Then those cells are further subdivided by which duration produced the strongest response, which means each group of cells is guaranteed to have a peak at those particular duration. The second issue is that the averages shown in Fig 4B and 4D may or may not be representative of the individual tuning curves. I'm not sure what the solution is to these issues. The decoder analysis avoids the issues with inclusion/classification, but it also doesn't say anything about how individual neurons are tuned and would probably work perfectly fine even if they don't have clear peaks. At a minimum, the parameters of the tuning curve (preferred delay, tuning width, monotonicity) should be quantified on an individual cell basis and then the proportion of cells that show evidence of monotonicity should be reported, rather than relying on an averaged tuning curve to draw conclusions about the population.

A broader weakness in the case for an abstract, dimensional representation of time is that it's not clear how stable the tuning curves are over time. The results show that there is tuning immediately after the stimulus turns off and immediately before the animal is supposed to respond, but how many cells retain the same tuning in that interval? How well does a decoder trained on the early interval generalize to the later part? Based on Figure 6, it looks like the firing rate vectors change quite a lot during the delay interval, so this is a very important point to address.

There is also an issue with the writing that I consider major because of how difficult it made it to understand the results. There is little framing in the introduction, and the order in which the results are presented is confusing. The introduction hypothesizes that NCL adopts similar solutions to mammalian brains for computations involving time intervals, but the specific similarities are not presented until the discussion. As a consequence, it is difficult to understand the significance and rationale for the two negative neural results presented immediately after the behavior (l. 245-288), especially for readers who may not be familiar with the types of "time cells" seen in the literature and the distinction between premotor ramping activity and a more abstract dimensional encoding of time. It was even more confusing to have the "Single unit correlates" section refer to Figure 3 but not find any examples of how the exponential or time-rescaling analyses work. It would be much easier to understand the logic of the paper if the ramping vs abstract hypotheses were presented in the introduction. Or, if the authors prefer a more inductive approach, then the results should be presented in a different order, with the tuning analysis followed by analyses that exclude alternative explanations.

Minor issues:

The methods for the neural recordings appear to be missing. Some specific points that need to be addressed: Were the neural probes implanted before or after the animals had learned the task? If after, how long did they animals have to recover before being tested? What specific probes were used? How was the probe verified to be in NCL? How were recording sites distributed within NCL? Was the animal tethered? How were spikes sorted, and what were the criteria for considering a unit to be well-isolated? How many behavioral sessions were used to collect neural data? Were the data pooled across multiple sessions? Were there criteria for stability of single units across trials?

The behavioral data are convincing but there seem to be a lot of lapse trials in the 6-s condition, where the animal responded immediately after the cue ended, but the authors never comment on this. Are these trials where the bird decided it was better to take the 4-s timeout than to wait 6 s for a reward? One might imagine this would be an optimal strategy if it allowed the bird to skip the longer trials in favor of the shorter ones. On a more practical note, were these trials included in

the Gaussian fits? If they were excluded, how? And if included, it seems like this would have a big impact on estimates of the standard deviation. In the neural analysis, it appears that the lapse trials are excluded, but a more precise term should be used to denote the remaining trials, as "correct" already has a clear definition that it is at odds with how it's being used.

It would be good to show what is happening during the pre-cue and cue period. It appears in some of the examples that there are already differences in firing rates when the cue turns off. At what point do these differences emerge in the trial?

The possibility of ramping activity in NCL is dismissed too easily. It's not clear how to add up all the percentages in l. 264 and l. 268, but it seems like ramping neurons could account for as much as 20% of the recorded population, which is small but could be indicative of the sort of mixed selectivity seen in mammalian prefrontal areas. It also seems like the simple exponential fit is a rather negatively biased approach to looking for ramping activity. The kind of oscillatory activity shown in the examples would lead to large errors even there are ramping trends, if the function being fitted is only exponential. This would lead to an underestimate of the proportion of neurons with ramping activity. An approach like Chebyshev regression that can accommodate both oscillatory and ramping trends would be more appropriate.

Examples of the exponential fit and rescaling analyses need to be shown.

l. 222: "across trials over for each duration" is a typo. Also, "binned" typically implies that spikes are assigned to non-overlapping intervals; "smoothing" is more accurate if there is a sliding window.

l. 296: how many / what proportion of neurons passed the criterion during both the early and late phase?

Figure 5: In panels A and B, what do the error bars represent? For a bootstrap test like this, standard error would not be appropriate, so please specify. In panels B and D, I believe the plot is showing the proportion of trials from each delay that were assigned to each delay by the classifier, in which case, the y-axis on the top plot should not read "Accuracy". What is the denominator - trials, resamples?

Figure 6: In panel 6, please indicate the direction of the trajectories and explain what the markers indicate.

Reviewer #3

(Remarks to the Author)

Review report for NCOMMS-24-50946

The authors trained two carrion crows on a delayed-response task to test whether the birds are able to estimate the duration of time periods and to study how time intervals are processed in the crows' associative endbrain NCL. The crows had to learn to delay their response to the instruction cue by a specific time interval (1.5, 3 or 6 s) instructed by the cue. The results show that both crows were able to successfully perform the behavioral task and that activity of a population of NCL neurons varied according to the instructed waiting time. The authors claim that these neurons are tuned to one of the three time intervals and that consequently these neurons must encode time intervals. While the question is interesting, the results are not very convincing and the main conclusion is not supported by the data.

A causal link between the finding that activity of NCL neurons varied with instructed waiting time and the conclusion that these neurons encode waiting time is completely lacking. The neurons' preference for specific waiting times can have many different causes unrelated to timing perception. The neurons' activity pattern could for instance reflect the bird's emotional state. Maybe the crow is happy when a short wait time is cued and sad or annoyed when a long wait time is cued. Neurons that respond stronger to short wait times than to long wait times could be tuned to the degree of happiness instead of time. Further, it was not tested whether the neural activity only varied according to the waiting time during the waiting period and not already before the waiting period. Response rates should be calculated for the pre-cue or cue period and compared with response rates during the waiting period. Such a test is crucial for showing that the change in neural activity is not an artefact but indeed due to the cued waiting time.

The number of animals used for the study is too small. Only two birds were trained and used for data collection. In addition, both birds were male. The authors should collect data in at least one additional crow, preferably a female bird, to exclude the possibility that the observations are sex-specific.

The neurophysiological methods are totally unclear. There is only one sentence in the methods section that vaguely describes the used recording system. It is neither mentioned what kind of electrode was used to record neural activity nor are any details about surgical procedures and analysis of raw neural data provided.

Line-specific comments:

Line 1: the title needs to be rewritten. There is no experimental proof for the assumption that neuron's in the crow's telencephalon signal impending time intervals.

Line 31: the word "time" needs to be removed.

Line 33-35: This sentence needs to be removed, since there is no proof that these neurons indeed encode waiting time.

Line 45 and 58: non-human animal species.

Line 69 – 74: a specific example on how time intervals play a role for natural behavior of carrion crows should be included.

Line 81: what is meant by "suggestive evidence"? Either evidence, or suggestion.

Line 83 - 84: the sentence needs to be rewritten. There is no behavioral representation in NCL.

Line 85 – 87: This sentence needs rewriting. A brain cannot adopt a solution, it can only develop in a way that it can perform certain computations.

Line 91: the age of the birds should be mentioned.

Line 114 – 116: a detailed description of neurophysiological methods needs to be provided, including surgical procedures, electrode types, etc. Was the electrode position fixed or changed daily?

Line 129: it should be mentioned which shape the grey border had? Was it of the same shape and size for all different cue stimuli?

Line 131: one of the two “that” can be removed.

Line 131 – 135: Why was the duration of the response period variable? The decision time should not depend on the cued wait time.

Line 137 and 139: the type of sound used as positive and as negative feedback needs to be specified.

Line 138: the type of reward needs to be specified.

Line 141 – 142: I think this protocol is problematic and could have affected the results, because the duration of the response period varied according to the cued stimulus. The crows were expected to respond faster when a short waiting period was cued than when a long waiting period was cued.

Line 142: what was displayed at the screen during the inter-trial interval?

Line 142 – 144: trial protocol is unclear. In which order were the different stimuli presented? Random order?

Line 144 – 145: How many trials per stimulus were on average included in one session?

Line 146: For how many days?

Line 162 – 164: was all data for each neuron collected within one session or over different sessions? Was the activity of neurons tuned to the waiting times only during the waiting period or already during the stimulus presentation period?

Line 166: what software was used for neural analysis? What was the average signal to noise ratio? How were spikes discriminated from background? What were the criteria for classifying a unit as a single-unit? What clustering algorithm was used?

Line 167: what base-criteria?

Line 169 – 172: the number of neurons classified as increasing and as decreasing should be mentioned in the results section.

Line 173: normalized to what?

Line 172 – 175: what was fitted to an exponential model? What software was used for this analysis step?

Line 185 – 189: This indicates that for trials with 1.5 s wait time, the two analysis windows largely overlapped. How did this affect the results? What exactly was the reason for choosing two analysis windows? Wouldn't it have been much easier to take the entire wait period for neural analysis?

Line 214: To which time period is referred here?

Line 218 – 220: why was the time period of cue presentation included in this part of the analysis but not in the others?

Line 232 – 234: why is the peak of the reaction time distribution expected to occur at a time before the minimum waiting time?

Line 243: the authors should comment on the fact that both crows responded very often before 1 s wait time during the 6s wait time trials. Why was that the case?

Line 246-247: recording sites should be indicated in Fig. 3. Have the recording sites been histologically verified at the end of the experiments?

Line 247-250: because the authors do not know what exactly the crow had perceived, this sentence should be written as a suggestion and not as a statement.

Line 250-253: I don't understand the reason for why a behavioral response that occurred prior to the appropriate wait time was considered correct. The explanation given here sounds completely random.

Line 260-262: Here, the percentage of neurons considered as “increasing” and as “decreasing” should clearly be stated.

Line 295-298: the numbers indicate that for 95 neurons a significant effect was found in both analysis windows. If this is the case, it should be mentioned.

Line 300-301: what about neurons that showed a significant effect in both analysis windows? Was the preferred wait duration equal for both analysis windows? If not, which duration was chosen as the preferred duration? How many neurons showed a difference in preferred wait time between the two analysis periods? How was the number of neurons for each preferred wait time distributed over the two crows? Were neurons preferring short duration and long durations, respectively, equally frequently found in both animals?

Line 312-314: in how many neurons the activity for the preferred wait duration was significantly different from the activity during the two non-preferred wait durations?

Line 387: why was the analysis performed only on a subset of neurons? The criteria to exclude data needs to be mentioned.

Line 359: why was the analysis performed only on a subset of neurons? The criteria to exclude data needs to be mentioned.

Line 459-460: I strongly disagree with that statement. Without any other timing-relevant evidence, such as ramping of activity with elapsed time, the authors can't just conclude that the variation of neural activity with different waiting periods is indicating that NCL neurons encode time. There are many alternative explanations for NCL neurons to vary their activity according to instructed wait time.

Line 591-592: why was it important that the wait and the go period were of the same duration?

Figure 2: it should be specified what time the x-axis label is referring to. Time relative onset of go-period? The bin-size of the histograms should be indicated in the figure caption. The dotted vertical line indicating the cued time interval can hardly be seen in the plot.

Figure 3: without any indication of the locations of recording sites, panel A is not very informative. Panel B and C should be swapped. According to the figure caption, panel B shows a neuron selective for 1.5 s, but in panel B, the red activity profiles demonstrate lowest firing rates. The grey shading is hardly visible in panels B-D. In the raster plot for the 6s wait time trials in panel D, the neuron seems to not spike at all during many subsequent trials, while there were some spikes in other trials. A complete lack of neural activity during some trials but not during others can be a sign of a systematic failure of the recording equipment. Was the lack of activity specific to the wait period or was this also seen during pre-cue and cue periods of the specific trials?

Figure 4: The authors may want to change the color pattern in this figure, to enable color-blind people access to the data. There is a typo in the Y-axis label in panel A and C.

Version 1:

Reviewer comments:

Reviewer #1

(Remarks to the Author)

The authors answered properly all the reviewers' and now the paper delivers a strong take home message.

Minor comments:

Please refer to the paper by Manuel Beiran et al., 2023 untitled: Parametric control of flexible timing through low-dimensional neural manifolds. This paper is important for the present study because using recurrent neural networks, they found that confining the dynamics to a low-dimensional subspace allowed tonic inputs to parametrically control the overall input-output transform, enabling generalization to novel inputs and adaptation to changing conditions. The tonic inputs they used are similar to your tonic interval tuned neurons.

Typo: Single-unit correlates of time estimation

Reviewer #2

(Remarks to the Author)

The revised manuscript from Johnston, Kirschhock, and Nieder is highly responsive to the reviewer comments. New analyses were added to address many of the questions raised, and extensive changes were made to the text and figures. Overall, the paper is much stronger, but there are still a few remaining and new issues.

The authors added a new analysis based on an SVM classifier (lines 267-286 and 503-511) to test the alternative hypothesis that NCL neurons encode elapsed time (as opposed to intended waiting time, which is their main hypothesis). I think the analysis is basically sound, but the description in the Methods is quite confusing, and the motivation could be explained more. Because SVM classifiers are employed in different ways throughout the manuscript, one helpful clarification would be to explicitly state what the features and categories are for each analysis. The underlying question seems to be whether the average firing rate of a neuron varies enough over time and is consistent enough across trials for an SVM classifier to decode what proportion of the interval has elapsed. As the classifier makes no assumptions about how the rate varies, it's a more general analysis than the linear/exponential/sigmoidal fit or the time-rescaling analysis and so it would make more sense coming first in this section rather than last. Some important details seem to be missing: how does the classifier performance vary across the population, and how are the data shown in Figure 5 related to the output of the model? Are the median and interquartile ranges across neurons, bootstrap replicates, or validation folds?

The authors adequately addressed my concern about selection bias in their rebuttal and by showing more data for individual neurons (Figure 3E,F is very helpful here). For tuning curve width, I am not sure that fitting a Gaussian function to three data points is the best approach, because there may be a lot of error and numerical instability in the nonlinear fitting procedure. Calculating the sample standard deviation across the three points should be more numerically stable.

The analysis of error trials is interesting and important but also somewhat difficult to follow in the methods and results. Why were the "late-error" but not the "early-error" trials analyzed? Looking at the response time histograms, it seems like there should be quite a few early-error trials for each category but almost no late-error trials (e.g., there are no blue bars after 6000 ms). Are these trials where the bird doesn't respond at all? In that case, the failure of the classifier could reflect the birds not being engaged in the task rather than an error prospectively encoding the interval.

Minor comments:

l. 31: "the executive telencephalon of male crows" implies that female crows might have a different executive telencephalon, which is probably not intended; please reword.

l. 205: Criteria used to determine if clusters were single units (e.g. signal-to-noise ratio, some index of cluster separation from noise, proportion of refractory period violations in the autocorrelations) is still missing.

l. 214: "trials" instead of "responses" would be clearer and more consistent with rest of the paper.

l. 412: Why was the mean of the Gaussian manually specified instead of being fit from the data?

l. 461: Recommend using past tense throughout Results.

l. 465: monotonicity -> monotonic

l. 466, 468: "monotonous" means boring. I think you mean "monotonic"

l. 485: "scalability of neuronal activity" could mean rescaling in time or in amplitude, please be specific.

Figure 3E,F: the plots show about 260 neurons, which doesn't correspond to the total number of units (409) or the number of time neurons (238). Please specify what neurons are shown. Also, what is 0 in the normalization? The minimum rate for each neuron, or 0 Hz?

l. 724: arcopallium, not archipallium

l. 732: with apologies for not catching this earlier, I don't think it's accurate to say that the avian pallium doesn't have layers,

it's just that they're not as distinct, universal, or formed in the same way as in the mammalian isocortex.

Reviewer #3

(Remarks to the Author)

I thank the authors for thoroughly revising their manuscript. Now, it reads like a very interesting and convincing story. Especially the comparison of neural activity between correct and incorrect trials and the new result on a possible sequential representation of time by ensembles of NCL neurons are exciting. All my comments were fully addressed and the manuscript can now be accepted for publication.

I only have a couple of minor comments:

Line 31: the word "male" should be moved to line 27 (We trained two male crows ...) otherwise it reads as if only male crows possess an executive telencephalon.

Line 184: please specify how many microdrives were implanted per bird and per hemisphere, and how many electrodes contained each microdrive?

Line 191: By what distance was the electrode moved on average each day? Were all electrodes in one drive moved simultaneously or independently?

Line 558 – 560: Could you please show an example of a response of one neuron to a specific target time during a correct trial and during an incorrect trial? I would be interested in seeing whether there are already differences in activity at the single neuron level.

Line 644: I suggest to rephrase this headline to make it not sound so negative. Maybe: "Sequential encoding of time by NCL neuron ensembles"

Figure 4 A and D are not mentioned in the text.

Version 2:

Reviewer comments:

Reviewer #2

(Remarks to the Author)

The authors have adequately addressed all of my concerns, and the paper is ready for publication. This is a strong contribution to the field, and the methodology is sound.

Reviewer #3

(Remarks to the Author)

I thank the authors for addressing all my remaining comments and congratulate them on a great paper.

Response to reviewers

Nature Communications manuscript NCOMMS-24-50946

Reviewer #1 (Remarks to the Author):

This paper investigates the neural correlates of interval timing in the nidopallium caudolaterale of crows executing a single interval production task using a range of durations between 1500 to 6000ms. The animals were able to reproduce the three target intervals with a precision that followed the scalar property of interval timing. The neurons showed no ramping activity nor clear temporal scaling. In contrast, a large population of cells was tuned to one of the three tested intervals, and accurate target duration classification was observed using their activity. Finally, the neural population trajectories showed distinctive dynamics for each target interval.

This is an interesting and well performed study, providing novel information regarding the neural signals associated with timing an action in the range of seconds in the pallium of crows. The behavior of the crows is very robust, and neurophysiology shows that neurons encode time using an interval tuning signal. The idea of using different visual cues to trigger the same timing performance is very clever. I have four main concerns. First, it is not clear how the authors computed the reaction times, they start to count after the cue offset? If this is the case the right tail of the RT distributions includes incorrect trials. Please make sure this is clear. In addition, it would be nice to know what happened with the responses of tuned neurons during error trials. Second, it would be interesting to determine whether the activity of timing neurons can decode elapsed time from the binned activity of neurons (Merchant and Averbeck, 2017). Third, I recommend that the authors perform analyses to determine whether the timing neurons form neural sequences, where neurons show a progressive pattern of activation that fills the time intervals (Crowe et al., J Neurosci; Gouvea et al., 2015 eLife; Zhou, et al., 2022. PLOS Computational Biology; Gamez et al., 2019 Plos Biol). Finally, it is important to measure the changes in amplitude, the speed and the final position of the neural trajectories for the 3 target durations. It has been shown that elapsed time is encoded in the final position of the trajectories (Sohn et al., Neuron 2019; Bi and Zhou, 2020 PNAS; Merchant and Perez, Nature machine intelligence 2020), whereas the speed and amplitude can encode prediction of future events (Wang et al., 2018 Nat Neurosci; Betancourt et al., Cell Rep 2023).

We thank the reviewer for their thoughtful feedback on our manuscript. We have adopted their suggestions for further analyses. Overall, the results from the analyses provide further support for conclusions that the NCL is important for timing behaviour and we feel this has greatly improved our manuscript.

First, we clarified our definitions of correct and incorrect trials on **lines 213–217**. After clarifying our definitions, we re-ran our analyses so that we could look at both the correct and incorrect trials separately. We found that neuronal activity during error trials is fundamentally different from correct trials, providing evidence of functional significance of the neuronal data. Changes can be found on **lines 308–314** for methods and **548–559** for results, as well as in **new Figure 7**.

Second, we binned neural activity of time neurons and trained and tested linear multiclass support vector machine (SVM) classifiers to test whether elapsed time could be decoded at the single unit level. We found that the classifier performance was highly accurate in predicting elapsed time of binned activity, suggesting some level of encoding of elapsed time at the single unit level. These updates can be found on **lines 265–284** and **501–509** (methods and results respectively) and in **new Figure 5**.

Third, we generated surface plots to determine whether the time neurons form neural sequences. We found NCL neurons peaked at distinct times during the wait periods (rather than fire at a steady rate), suggesting that ensembles of time neurons form dynamic sequences over time. **Lines 237–241** and **450–452** (methods and results respectively) and **new Figure 3E–F**.

Fourth, we measured the amplitude and speed of the state-space trajectories. Amplitude was calculated as the Euclidean distance of each point in the neural (3D) trajectory from the origin (i.e., the first point) of the trajectory. Velocity was determined by calculating the change in position along the trajectory—specifically, the Euclidean distance between consecutive points—divided by the time step between them (in seconds). These changes can be found in **lines 392–397** and **614–617** for methods and for results, respectively, as well as in **new Figure 9C,D**.

Fifth, all suggested citations have been incorporated.

Minor comments.

It would be useful that the x-axis of figure 2 show the target interval and depict the means of the gaussian distributions.

We thank the reviewer for their suggestion. The target intervals in Figure 2 have been updated. The means of the Gaussian distributions are shown in the **inset of Figure 2**. To avoid overcrowding the main graph, we chose to present this information in the inset, keeping the overall figure clear and interpretable. We hope this format effectively conveys the data while maintaining visual simplicity.

Please also report the number of neurons with significant main effect of stimulus protocol and the interaction target duration x stimulus protocol.

Thank you for raising this important point. We have included these details in the revised manuscript (**lines 434–441**).

It would be nice to have figure showing the limited scaling in the neural responses. Did you find an increase of the width of duration tuning as a function of the interval? Did the authors find differences between crow individuals in duration tuning, the decoding and neural trajectories?

We thank the reviewer for highlighting these important aspects previously not included in the manuscript. We have included new analyses, a new main figure, and three supplementary figures to address these points.

In **new Supplementary Figures 1**, we show that there is no difference in the tuning behavior of single neurons across the two crows (**line 460**).

In **new Supplementary Figures 3**, we show that neuronal population decoding of target duration is indifferent for both crows (**line 533**).

We therefore conclude that it is appropriate to collapse further analyses across the two crows.

We also included a **new Supplementary Figure 2** showing the limited scaling of the neuronal responses (**lines 492**).

In the main text we now provide **new Figure 4C,F** demonstrating the increasing tuning width with increasing in target durations (**lines 460–462**).

Why the Euclidean distance in Figure 6B showed a peak before the time estimation onset?

Thank you for this insightful observation. Time point 0 marks the onset of the wait period, which corresponds to the onset of the time estimation phase. However, in the 600 ms before the time estimation onset, the cues were presented. The display of these cues may have altered the neuronal responses and/or allowed the crows to retrieve the associated time period from long-term memory and plan for the waiting time. These factors likely explain why the Euclidean distances show a peak before the time estimation onset. This information has now been added on **lines 609–611** and to the legend of **Figure 9B**.

In the discussion, please consider that the paper of Merchant et al., 2011 PNAS describes the properties of ramping neurons in medial premotor areas, while the paper Merchant et al., 2013 J Neuroscience deals with interval tuning during single interval or rhythmic time production tasks.

Thank you for highlighting this distinction. We have carefully revisited both studies and incorporated them appropriately into the introduction and discussion, which has significantly improved the clarity and structure of our arguments. We appreciate the reviewer's attention to this detail.

Typos

line 82 ,2019) However, t

Thank you; corrected

Figure 4a: Nuerons

Corrected.

Reviewer #2 (Remarks to the Author):

This study by Johnston, Kirschhock, and Nieder examines how time is represented in the brains of carrion crows performing a well-trained cued delay task. The neural recordings were obtained from the caudolateral nidopallium, which is thought to be the avian equivalent of prefrontal cortex in mammals. The behavioral performance was consistent with a magnitude-based representation of time, in that the variance of the errors scaled with the duration of the interval. While the birds were waiting, a subset of NCL neurons exhibited tuning for specific delay durations, and the population activity jointly encoded the cued delay time with a high degree of accuracy, which suggests that NCL prospectively encodes delay time, possibly as a dimensional quantity. This behavior is consistent with what has been seen in primate prefrontal cortex, and in contrast to the ramping activity seen in downstream motor areas.

A finding that the crow brain represents time similarly to the brains of primates is important because of how radically different the neural architectures are of these widely diverged species. For the most part, the data supporting this claim are convincing. The number of animals and number of units are appropriate given the difficulty of the experiments, and the experimental and analytical methods are generally sound. The manuscript is well-written and clear except for a significant structural issue described below. Most of the weaknesses are minor and should be easy to address in a revision. However, the results as presented fall short on some major points.

Scalar expectancy theory makes two predictions about the neural representation of magnitudes. First, it should be abstract, independent of the physical characteristics of the stimuli that cue the behavior. This is very clearly supported by the data in this study: there are a large number of time cells that encode the duration signified by the cue without depending on whether the duration was cued with color or shape, and a classifier trained on population activity during one type of cue successfully generalized to the other type of cue. Second, as the authors note around line 317, the representation should be ordered and dimensional, with tuning curves that have a clear peak on the time scale and fall off more or less monotonically in either direction. This is the point that I am not sure has been convincingly demonstrated, for three main reasons.

The first is that time cells are included and classified based on measures that are later used as dependent measures to support the claim that the tuning curves have a particular shape. The inclusion criterion, a significant main effect of target time in a two-way ANOVA, is going to select for neurons that have a peaked tuning curve. Then those cells are further subdivided by which duration produced the strongest response, which means each group of cells is guaranteed to have a peak at those particular duration. The second issue is that the averages shown in Fig 4B and 4D may or may not be representative of the individual tuning curves. I'm not sure what the solution is to these issues. The decoder analysis avoids the issues with inclusion/classification, but it also doesn't say anything about how individual neurons are tuned and would probably work perfectly fine even if they don't have clear peaks. At a minimum, the parameters of the tuning curve (preferred delay, tuning curve width, monotonicity) should be quantified on an individual cell basis and then the proportion of cells that show evidence of monotonicity should be reported, rather than relying on an averaged tuning curve to draw conclusions about the population.

We appreciate the reviewer's concern regarding the potential for a biased selection in our inclusion criteria. However, we have followed standard practices in the field to identify tuned neurons, not only time-tuned neurons, but tuned neurons in general, across all sensory, motor and cognitive systems. For instance, we and others use exactly the same method to select numerosity-tuned neurons in humans, monkeys, and crows; all of these studies have been published in respected journals after peer-review. We have taken care to present the data as objectively and transparently as possible, and using an ANOVA is a most objective way of selecting tuned neurons. The ANOVA is not specific to peak tuning; any neuron that shows a significantly higher firing rate on of the estimation times compared to the other is included. The ANOVA could have found purely

monotonically increasing (or decreasing) neurons that would show no peak tuning curve. The ANOVA is therefore not selecting for particular tuning curve shapes.

We appreciate the reviewer's suggestion to quantify the tuning parameters, such as the tuning width and monotonicity (preferred tuning is already reported) on an individual cell basis. Accordingly, in the revised manuscript we fit and plot the width of Gauss curves and monotonicity of 1500 ms and 6000 ms neurons using a Jonckheere-Terpstra test (**new Figure 4C,F; lines 230–236 and 460–467**). Overall, we found a trend of an increasing tuning width with an increase in target duration but very few neurons that showed non-monotonicity. Additionally, in a supplementary figure, we now show the tuning curves of each crow separately (**new Supplementary Figure 1; line 459**).

A broader weakness in the case for an abstract, dimensional representation of time is that it's not clear how stable the tuning curves are over time. The results show that there is tuning immediately after the stimulus turns off and immediately before the animal is supposed to respond, but how many cells retain the same tuning in that interval? How well does a decoder trained on the early interval generalize to the later part? Based on Figure 6, it looks like the firing rate vectors change quite a lot during the delay interval, so this is a very important point to address.

We thank the reviewer for highlighting this important point, as the stability of neuronal activity over time is a critical aspect that we had previously overlooked. To address this, we conducted three new analyses to assess the stability and persistence of the temporal representation. All three analyses provide further support of our conclusions and make the manuscript stronger.

First, we performed a percent explained variance (PEV) analysis, which quantifies the amount of information about different task factors carried by neuronal populations. We found that information related to cue properties declined shortly after cue offset, while representation of the target duration remained highly persistent throughout the delay interval (**new Figure 8A**). This suggests that the target duration was the most prominently encoded and behaviorally relevant factor (**lines 318–340 and 562–570**, methods and results, respectively).

Second, we trained support vector machine (SVM) classifiers on the firing rates from a given time point and tested them on firing rates from any other time points in a new trial. The classifier maintained high accuracy throughout both the cue and wait periods (**new Figure 8B**), indicating that the target duration is encoded not only during the cue presentation but also while the animal is actively estimating time during the wait period (**lines 341–365 and 571–586**, methods and results, respectively).

Third, we split the wait period into equal-sized bins and trained SVM classifiers on firing rates from individual bins using a subset of trials, then tested them on firing rates from previously unseen trials. This analysis revealed a stable encoding of time across the wait period (**new Figure 8C**), further supporting the robustness of the temporal representation (**lines 366–378 and 587–595**, methods and results, respectively).

There is also an issue with the writing that I consider major because of how difficult it made it to understand the results. There is little framing in the introduction, and the order in which the results are presented is confusing. The introduction hypothesizes that NCL adopts similar solutions to mammalian brains for computations involving time intervals, but the specific similarities are not presented until the discussion. As a consequence, it is difficult to understand the significance and rationale for the two negative neural results presented immediately after the behavior (l. 245-288), especially for readers who may not be familiar with the types of "time cells" seen in the literature and the distinction between premotor ramping activity and a more abstract dimensional encoding of time. It was even more confusing to have the "Single unit correlates" section refer to Figure 3 but not find any examples of how the exponential or time-rescaling analyses work. It would be much easier to understand the logic of the paper if the ramping vs abstract hypotheses were presented in the introduction. Or, if the authors prefer a more inductive approach, then the results should be presented in a different order, with the tuning analysis followed by analyses that exclude alternative explanations.

We appreciate the reviewer's feedback and improved the framing and organisation of the manuscript. In response, we have revised the introduction to present the competing hypotheses

regarding categorical representations of elapsed time versus continuous ramping representations in the brain (**lines 54–66**). Additionally, we have restructured the methods and results to follow a more logical progression, beginning with single-unit analyses and expanding to the population level to systematically evaluate evidence for each hypothesis in the avian brain. We believe this revised structure enhances the manuscript's readability and better guides the reader through the rationale and significance of each analysis.

Minor issues:

The methods for the neural recordings appear to be missing. Some specific points that need to be addressed: Were the neural probes implanted before or after the animals had learned the task? If after, how long did they animals have to recover before being tested? What specific probes were used? How was the probe verified to be in NCL? How were recording sites distributed within NCL? Was the animal tethered? How were spikes sorted, and what were the criteria for considering a unit to be well-isolated? How many behavioral sessions were used to collect neural data? Were the data pooled across multiple sessions? Were there criteria for stability of single units across trials?

We apologise to the reviewer for the oversight in not including more details regarding the surgical and neuronal recording procedures. We have updated the manuscript to include a dedicated section ("Surgery and neuronal recordings"; **lines 174–203**) which outlines the details of these procedures. We have also provided the remaining details throughout the Methods and Results sections where appropriate.

In alignment with the Three Rs principle to reduce the number of animals used in research, we use the same crows across multiple experiments and do not sacrifice them after each study. While post-experiment sacrifice and histological processing would enable precise confirmation of neural recording locations, exact locations were not obtained for this specific experiment. However, the implant coordinates were based on previous work (Veit, Hartmann, and Nieder, 2014), where locations were verified post-experimentation (**lines 184–187**). Given this, we are confident that the recording site in the current study is in the NCL.

The behavioral data are convincing but there seem to be a lot of lapse trials in the 6-s condition, where the animal responded immediately after the cue ended, but the authors never comment on this. Are these trials where the bird decided it was better to take the 4-s timeout than to wait 6 s for a reward? One might imagine this would be an optimal strategy if it allowed the bird to skip the longer trials in favor of the shorter ones. On a more practical note, were these trials included in the Gaussian fits? If they were excluded, how? And if included, it seems like this would have a big impact on estimates of the standard deviation. In the neural analysis, it appears that the lapse trials are excluded, but a more precise term should be used to denote the remaining trials, as "correct" already has a clear definition that it is at odds with how it's being used.

We thank the reviewer for pointing out these observations. In response, we have added a discussion of the frequent early responses observed during 6000 ms trials. We agree that part of the behaviour may represent a strategic choice by the crows to maximize their overall efficiency in the task. We have incorporated this explanation into the discussion section of the revised manuscript (**lines 631–640**). Regarding the practical aspect of the Gaussian fits, we accounted for the lapse trials in the 6-s condition by excluding the first 400 ms of response times from these trials. This exclusion was based on the observation that responses occurring immediately after the cue were not representative of genuine interval estimates but rather reflected an opt-out strategy by the birds. This detail has been included in the revised manuscript (**lines 404–411**).

Additionally, we have clarified the definitions of "correct" trials in the manuscript to ensure consistency and accuracy in terminology (**lines 212–219**). Additionally, we have included further analyses comparing neuronal data from correct trials and error trials to provide a clearer understanding of the relationship between neural activity and behavioural performance (**new Figure 7**).

It would be good to show what is happening during the pre-cue and cue period. It appears in some of the examples that there are already differences in firing rates when the cue turns off. At what point do these differences emerge in the trial?

We thank the reviewer for this insightful observation. To address this, we have included additional population analyses that incorporate activity from the pre-cue and cue periods.

Notably, the new percent explained variance (PEV) analysis reveals that information regarding the target duration was negligible prior to cue onset, whereas information regarding the cue properties was high after the cue onset but declines shortly after cue offset. This suggests that while cue-related information is transient, the target duration remains the most prominently encoded and behaviorally relevant factor throughout the wait period. These changes can be found on **lines 318–340** and **561–570** (methods and results, respectively) and in **new Figure 8A**.

Furthermore, the **new SVM classifier analysis (lines 341–365 and 571–586, methods and results, respectively; new Figure 8B)** shows that classification accuracy was at chance during the pre-cue period, indicating the absence of task-related information at this stage. However, accuracy increases significantly during the cue presentation and remains high throughout the wait period. This finding demonstrates that the wait duration is robustly encoded both during the cue presentation and while the crows are actively estimating time.

The possibility of ramping activity in NCL is dismissed too easily. It's not clear how to add up all the percentages in l. 264 and l. 268, but it seems like ramping neurons could account for as much as 20% of the recorded population, which is small but could be indicative of the sort of mixed selectivity seen in mammalian prefrontal areas. It also seems like the simple exponential fit is a rather negatively biased approach to looking for ramping activity. The kind of oscillatory activity shown in the examples would lead to large errors even there are ramping trends, if the function being fitted is only exponential. This would lead to an underestimate of the proportion of neurons with ramping activity. An approach like Chebyshev regression that can accommodate both oscillatory and ramping trends would be more appropriate.

We thank the reviewer for raising this important point. In response, we have updated our analyses to provide a clearer and more intuitive assessment for the reader. To address this, we redefined the criteria for identifying neurons with increasing activity and fit each neuron with three models: linear, exponential, and sigmoidal. We then evaluated the goodness of fit (r^2) for each model against a distribution of r^2 values obtained from 100 shuffled label fits to determine statistical significance. From these analyses, we found that only 20 neurons (4.89%) showed a significantly better fit for a linear model, 21 neurons (5.89%) for an exponential model, and three neurons (0.73%) for a sigmoidal model. Importantly, we observed that the neurons with significant linear fits also showed significant exponential fits, indicating that these populations overlapped entirely. Thus, the new analyses continue to support the notion that very few single units in the NCL exhibit ramping activity to encode the passage of time (**lines 244–255 and 472–481, methods and results, respectively**).

Examples of the exponential fit and rescaling analyses need to be shown.

We thank the reviewer for this suggestion. We have updated the fitting analyses (see **response to previous comment**), and the new results are not suitable for direct visualization. However, we have now included a figure in the supplementary material to illustrate the rescaling analyses, as requested (**new Supplementary Figure 2**).

l. 222: "across trials over for each duration" is a typo. Also, "binned" typically implies that spikes are assigned to non-overlapping intervals; "smoothing" is more accurate if there is a sliding window.

We thank the reviewer for highlighting these errors. The manuscript has been updated accordingly.

l. 296: how many / what proportion of neurons passed the criterion during both the early and late phase?

The revised manuscript now includes the number of neurons that were selective during both the cue offset and response onset windows of analysis (**lines 434–441**)

Figure 5: In panels A and B, what do the error bars represent? For a bootstrap test like this,

standard error would not be appropriate, so please specify. In panels B and D, I believe the plot is showing the proportion of trials from each delay that were assigned to each delay by the classifier, in which case, the y-axis on the top plot should not read "Accuracy". What is the denominator - trials, resamples?

We thank the reviewer for their careful observation. We confirm that the error bars in **now Figure 6, panels A and B**, represent the standard error of the mean (SEM), and we are confident that this is an appropriate choice for these analyses as error bars represent the variance across repetitions (and we have used such displays before). Additionally, we have updated the y-axis labels in panels B and D to read "% decoded" to clarify the interpretation of the plots.

Figure 6: In panel 6, please indicate the direction of the trajectories and explain what the markers indicate.

We apologise for the oversight and thank the reviewer for pointing this out. This figure (**now Figure 9A**) and caption have been updated to include numbers related to trial phases and thus indicating the direction of the trajectories for each trial phase.

Reviewer #3 (Remarks to the Author):

The authors trained two carrion crows on a delayed-response task to test whether the birds are able to estimate the duration of time periods and to study how time intervals are processed in the crows'; associative endbrain NCL. The crows had to learn to delay their response to the instruction cue by a specific time interval (1.5, 3 or 6 s) instructed by the cue. The results show that both crows were able to successfully perform the behavioral task and that activity of a population of NCL neurons varied according to the instructed waiting time. The authors claim that these neurons are tuned to one of the three time intervals and that consequently these neurons must encode time intervals. While the question is interesting, the results are not very convincing and the main conclusion is not supported by the data.

A causal link between the finding that activity of NCL neurons varied with instructed waiting time and the conclusion that these neurons encode waiting time is completely lacking. The neurons' preference for specific waiting times can have many different causes unrelated to timing perception. The neurons' activity pattern could for instance reflect the bird's emotional state. Maybe the crow is happy when a short wait time is cued and sad or annoyed when a long wait time is cued. Neurons that respond stronger to short wait times than to long wait times could be tuned to the degree of happiness instead of time.

We acknowledge the reviewer's concern. Just as most other studies that report behaviourally-relevant neuronal responses, including time-related activity, we rely on correlational measures, and we clearly state this now in the new title **"A neuronal correlate for impending time interval estimation in the crow's telencephalon"** and throughout the manuscript. Of course, we agree with the reviewer that measuring the consequences on time estimation by manipulating NCL would be a great asset. At moment, a lack of connectivity information of the NCL and the complex behavioural deficits after perturbation of the NCL prevent such a step.

However, even without causal manipulation, our recordings argue for behavioral relevance of the neurons. First, The neurons are tuned to all three different time intervals, which argues against the explanation put forth by the reviewer that they might reflect the crows' emotional state (such as "happiness" or "sadness"), which according to the reviewers argumentation, would be strongest only for short intervals. Secondly, and more importantly, we now show that the neurons react very differently in error trials, when the crows make mistakes. We re-ran our analyses so that we could look at both the correct and incorrect trials separately. We found that neuronal activity during error trials is fundamentally different from correct trials, providing evidence of functional significance of the neuronal data. This can be found on **lines 308–317** for methods and **548–559** for results, as well as in **new Figure 7**. In other words, if the neurons do not encode their proper time intervals, the crows are prone of making errors.

We now explicitly discuss the question of behavioral relevance and acknowledge the need for causal manipulation as a limitation of the current on **lines 714–727**.

Further, it was not tested whether the neural activity only varied according to the waiting time during the waiting period and not already before the waiting period. Response rates should be calculated for the pre-cue or cue period and compared with response rates during the waiting period. Such a test is crucial for showing that the change in neural activity is not an artefact but indeed due to the cued waiting time.

We thank the reviewer for this insightful observation. To address this, we have included additional population analyses that incorporate activity from the pre-cue and cue periods.

First, the new percent explained variance (PEV) analysis reveals that information regarding the target duration was negligible prior to cue onset. In contrast, information regarding the cue properties was high after the cue onset but declines shortly after cue offset. However, the target duration remains the most prominently encoded and behaviorally relevant factor throughout the wait period. These changes can be found on **lines 318–340** and **561–570** (methods and results, respectively) and in **new Figure 8A**.

Second, the **new SVM classifier analysis** (**lines 341–365** and **571–586**, methods and results, respectively; **new Figure 8B**) shows that classification accuracy was at chance during the pre-cue

period before cue presentation, indicating the absence of task-related information at this stage. However, accuracy increases significantly during the cue presentation and remains high throughout the wait period. This finding demonstrates that the wait duration is robustly encoded both during the cue presentation and while the crows are actively estimating time. Both analyses confirm that the change in neural activity is not an artefact but indeed due to the cued waiting time.

The number of animals used for the study is too small. Only two birds were trained and used for data collection. In addition, both birds were male. The authors should collect data in at least one additional crow, preferably a female bird, to exclude the possibility that the observations are sex-specific.

We appreciate the reviewer's attention to these important details. Regarding potential sex differences, we have no reason to believe that there are any differences in timing between male and female crows, particularly at the neural level. However, to ensure clarity for readers, we have specified "male" in the abstract to indicate that our study was conducted exclusively with male crows.

Concerning the number of subjects, we believe that two crows are appropriate for this type of experiment, especially since we did not observe any differences between them in the measured variables. Adding a third female crow would not provide additional insights regarding the potential presence or absence of sex-specific effects, as the number of subjects is still too low to address such questions.

Following the principles of the 3Rs (replacement, reduction, refinement), we are required, according to the animal welfare act and our animal experimentation license, to use the minimum number of subjects necessary. Two subjects is the minimum number required to achieve reliable results and minimize animal use. Due to the high training demands, using two animals is standard practice for this type of study. Over the years, we have published in renowned peer-reviewed journals, including *Nature Communications*, with two crows per study (e.g., Veit L, Nieder A, Nat Commun., 2013; Veit L, Pidpruzhnykova G, Nieder A, Proc Natl Acad Sci U S A, 2015; Nieder A, Ditz HM, Nieder A, Nat Commun., 2020; Wagener L, Rinnert P, Science, 2020; Kirschhock ME, Nieder A, Nat Commun., 2022; Wagener L, Nieder A, Curr Biol., 2023, etc.).

The neurophysiological methods are totally unclear. There is only one sentence in the methods section that vaguely describes the used recording system. It is neither mentioned what kind of electrode was used to record neural activity nor are any details about surgical procedures and analysis of raw neural data provided.

We apologise to the reviewer for the oversight in not including more details regarding the surgical and neuronal recording procedures. We have updated the manuscript to include a new dedicated section ("***Surgery and neuronal recordings***"; lines 174–203) outlining the details of these procedures.

Line-specific comments:

Line 1: the title needs to be rewritten. There is no experimental proof for the assumption that neurons in the crows telencephalon signal impending time intervals.

We acknowledge that we are using a correlative measure to assess the neuronal mechanisms of time estimation and have adapted the title in response to the reviewer's comment. However, we maintain that single neuron recordings provide strong evidence for behaviorally-relevant neuronal activity by directly correlating neuronal firing patterns with specific behaviors in real time. Decades of research, including Nobel-winning work, for instance by Hubel and Wiesel on visual processing, have demonstrated the value of this approach. Single neuron recordings offer high temporal resolution, capturing rapid neuronal changes that align with behavioral events, thus providing insights into how neurons contribute to behavior. Neurons often show selective activity in response to specific stimuli, motor actions, or cognitive processes, and these recordings help identify how neurons encode behavioral features. This method reveals the role of individual neurons in processes like decision-making, motor control, and memory retrieval. While we agree that pairing these recordings with techniques like optogenetics can establish causality, we first need to rely on

correlative measures, such as single neuron recordings, in species like crows. Controlled perturbations like optogenetics are not yet feasible in crows, so we must start with this foundational approach before progressing to more complex methods. In summary, single neuron recordings offer a powerful and detailed method for understanding how neuronal activity underlies and supports behavior.

Line 31: the word “time” needs to be removed.

The revised manuscript has removed the word “time” for labelling neurons whose firing rates varied according to the impending waiting duration, each tuned to one of three specific waiting periods.

Line 33-35: This sentence needs to be removed, since there is no proof that these neurons indeed encode waiting time.

Please refer to our argument in favor of single-neuron recordings in our reply to specific comment, Line 1.

We have revised the sentence to soften the language and avoid implying definitive proof that these neurons encode waiting time. The updated sentence now uses more cautious phrasing, such as “reflected” and “suggesting”, to better convey that the findings constitute a correlative measure.

Line 45 and 58: non-human animal species.

The manuscript has been updated to include the terminology “non-human animal species” (line 43).

Line 69 & 74: a specific example on how time intervals play a role for natural behavior of carrion crows should be included.

Research on corvids has explored how various species use timing in foraging behaviours, though there have been no explicit studies examining how crows incorporate time in their natural behaviours. In closely related species, such as scrub jays, timing plays a critical role in foraging, such that these birds make decisions based on when food will be most accessible or nutritious (e.g., Clayton & Dickinson, 1998). This study is highlighted in the manuscript (lines 72–74).

Line 81: what is meant by “suggestive evidence”? Either evidence, or suggestion.

“Suggestive evidence” is replaced by “led to the hypothesis” (line 82).

Line 83 - 84: the sentence needs to be rewritten. There is no behavioral representation in NCL.

We appreciate the reviewer’s feedback and have revised the sentence accordingly (lines 84–89).

Line 85 & 87: This sentence needs rewriting. A brain cannot adopt a solution, it can only develop in a way that it can perform certain computations.

We appreciate the reviewer’s feedback and have revised the sentence accordingly (lines 87–89).

Line 91: the age of the birds should be mentioned.

We thank the reviewer for the suggestion. We have updated the manuscript to include the approximate age of the subjects at the time of recording (lines 92–93).

Line 114 & 116: a detailed description of neurophysiological methods needs to be provided, including surgical procedures, electrode types, etc. Was the electrode position fixed or changed daily?

We apologise to the reviewer for the oversight in not including more details regarding the surgical and neuronal recording procedures. We have updated the manuscript to include a new dedicated section (“*Surgery and neuronal recordings*”; lines 174–203) outlining the details of these procedures.

Line 129: it should be mentioned which shape the grey border had? Was it of the same shape and size for all different cue stimuli?

We thank the reviewer for highlighting this. The manuscript has been updated to clarify that each cue was surrounded by an identical grey square border (lines 132).

Line 131: one of the two “that” can be removed.

We thank the reviewer for highlighting this. The manuscript has been updated to remove the repeated word.

Line 131 & 135: Why was the duration of the response period variable? The decision time should not depend on the cued wait time.

We scaled the response window to accommodate greater variability in responses for larger target estimations, aligning with the principle of scalar expectancy. While it was possible to use a larger response window for shorter estimations, doing so risked crows disengaging from the task during the estimation period, potentially leading to inaccurate classifications of later responses as "correct." Ultimately, a larger window for shorter estimations was unnecessary, as crows made very few errors in these cases. The manuscript has been updated to clarify this point (**lines 137–139**).

Line 137 and 139: the type of sound used as positive and as negative feedback needs to be specified.

The sounds are now specified (**lines 141 and 145**).

Line 138: the type of reward needs to be specified.

We apologise to the reviewer for the oversight in not including the specific details of the reward received by the crows. The revised manuscript includes this detail (**lines 142–144**).

Line 141 & 142: I think this protocol is problematic and could have affected the results, because the duration of the response period varied according to the cued stimulus. The crows were expected to respond faster when a short waiting period was cued than when a long waiting period was cued.

In our protocol, we instructed the crows to wait at least a minimum amount of time, rather than aiming for precise accuracy to the target time. After this minimum wait, it was in the birds' best interest to respond as soon as possible to maximize reward efficiency. While we did implement variable response windows to accommodate larger variability in responses for longer target estimations (scalar expectancy), the crows were free to wait longer (up to the set limit) if desired. This design allowed flexibility in response timing from the crows while still maintaining our ability to assess their time estimations accurately.

Line 142: what was displayed at the screen during the inter-trial interval?

The inter-trial interval had nothing displayed on the screen. We have updated the manuscript to reflect this (**line 149**).

Line 142 & 144: trial protocol is unclear. In which order were the different stimuli presented? Random order?

Trial types (red, ring, blue, triangle, green, plus) were randomly intermixed in blocks of 600 correct trials (100 per stimulus). We have updated the manuscript to clarify this point (**lines 149–152**).

Line 144 & 145: How many trials per stimulus were on average included in one session?

We thank the reviewer for the suggestion. We have revised the “Behavioral Protocol” section of the manuscript to include the average number of each trial type attempted (both correct and incorrect) by each bird (**lines 153–159**).

Line 146: For how many days?

We thank the reviewer for the suggestion. We have updated the “Behavioral protocol section” in the manuscript to include the number of days/sessions each crow completed (**lines 158–159**).

Line 162 & 164: was all data for each neuron collected within one session or over different sessions? Was the activity of neurons tuned to the waiting times only during the waiting period or already during the stimulus presentation period?

We thank the reviewer for their question. Each day, electrode adjustments were made until a clear neuronal signal (with at least a 3:1 signal-to-noise ratio) was detected on at least one channel. As

a result, each neuron was recorded during a single session. This detail has now been added to the updated manuscript (**lines 189–190**).

Our primary focus in this study was the wait period, and as such, we specifically examined neural tuning during this time. However, to address this point, we have now included a percent explained variance (PEV) analysis in the updated manuscript, which demonstrates that target duration was also encoded during the stimulus presentation period. These changes can be found on **lines 318–340** and **561–570** (methods and results, respectively) and in **new Figure 8**.

Line 166: what software was used for neural analysis? What was the average signal to noise ratio? How were spikes discriminated from background? What were the criteria for classifying a unit as a single-unit? What clustering algorithm was used?

We apologise to the reviewer for the oversight in not including more details regarding the surgical and neuronal recording procedures. We have updated the manuscript to include a new dedicated section ("***Surgery and neuronal recordings***"; **lines 174–203**) outlining the details of these procedures.

Line 167: what base-criteria?

We apologise for our unclear wording. We have updated the terminology in the manuscript such that we now refer to the "base criteria" as "the aforementioned inclusion criteria" (**line 245**).

Line 169 & 172: the number of neurons classified as increasing and as decreasing should be mentioned in the results section.

We thank the reviewer for pointing this out. In response to comments from another reviewer regarding this set of analyses, we conducted an alternative analysis focusing specifically on increasing neurons. As part of these revisions, we have now included the number of increasing neurons in the results section of the manuscript (**line 250, lines 475**).

Line 173: normalized to what?

As with the previous comment, this comment is no longer applicable as we conducted an alternative analysis focusing specifically on increasing neurons. However, we have included the details for normalisation in the revised manuscript (**lines 251–252**).

Line 172 & 175: what was fitted to an exponential model? What software was used for this analysis step?

All neuronal analyses were conducted in MATLAB (R2024b) using custom scripts. We have updated the manuscript to clarify this point (**lines 206**). While the neural data was initially fitted to an exponential model, this analysis is no longer directly relevant as we have now performed alternative analyses, as discussed in response to a previous comment (**line 252, lines 471–481**).

Line 185 & 189: This indicates that for trials with 1.5 s wait time, the two analysis windows largely overlapped. How did this affect the results? What exactly was the reason for choosing two analysis windows? Wouldn't it have been much easier to take the entire wait period for neural analysis?

We acknowledge that there is significant overlap between the analysis windows for the 1500 ms trials. While we would have preferred to use a single standard analysis window, this was not possible for any of the three wait durations due to variability in time estimations across trials (i.e., not all trials were of the same length). To address this, we used two windows of 1200 ms, which allowed us to capture the maximum amount of data while maintaining consistency across the three wait durations. Importantly, 1200 ms was the minimum duration required for a trial to be considered "correct," ensuring that our analyses focused on behaviourally relevant time periods. This approach enabled us to examine increases in neuronal firing across a trial to identify potential correlates of elapsed time, as we needed to include both the start and end of the wait period for this purpose.

Line 214: To which time period is referred here?

We thank the reviewer for their question. In this context, "each point in time" refers to each discrete moment of time throughout the trial period. Specifically, this includes the period from cue onset

through the instructed waiting time, allowing us to observe how neuronal population activity evolves over the entire time course of the trial (line 381–382).

Line 218 & 220: why was the time period of cue presentation included in this part of the analysis but not in the others?

We thank the reviewer for this question. The majority of the manuscript focuses on the wait period, which is why the pre-cue and cue periods were not included in earlier population analyses. However, in this particular analysis, our goal was to examine population changes over time rather than directly measuring correlates of elapsed time. This is why the cue presentation period was included here for the first time. In the revised manuscript provide two additional analyses—the PEV analysis and the SVM classifier analysis—that incorporate both the pre-cue and cue periods. These results offer a more comprehensive view of how neural activity evolves across different task phases. These changes can be found on lines 318–340 and 562–570 (methods and results, respectively) and in new Figure 8.

Line 232 & 234: why is the peak of the reaction time distribution expected to occur at a time before the minimum waiting time?

We thank the reviewer for their comment. We believe there may have been a misreading of our original wording. The sentence intended to convey that the peak of the reaction time distribution is expected to occur *just beyond* the minimum waiting time, not before it. To improve clarity, we have rephrased the sentence in the manuscript (line 406).

Line 243: the authors should comment on the fact that both crows responded very often before 1 s wait time during the 6s wait time trials. Why was that the case?

We thank the reviewer for highlighting this important observation. In response, we have added a discussion of the early responses observed during the 6000 ms trials. Specifically, we noted that both crows frequently responded within the first second of the wait period. We believe this behaviour may reflect a strategic decision by the crows to optimise their overall reward rate. By aborting the longer trials early and accepting the 4-second timeout, the crows could increase the likelihood of encountering a shorter trial (1500 ms or 3000 ms) in the next randomized sequence, reducing the overall wait time for a potential reward. This explanation has been incorporated into the discussion section of the revised manuscript (lines 631–640)

Line 246-247: recording sites should be indicated in Fig. 3. Have the recording sites been histologically verified at the end of the experiments?

In alignment with the 3Rs principle to reduce the number of animals used in research, we use the same crows across multiple experiments and do not sacrifice them after each study. While post-experiment sacrifice and histological processing would enable precise confirmation of neural recording locations, exact locations were not obtained for this specific experiment. However, the implant coordinates were based on previous work (Veit, Hartmann, and Nieder, 2014), where locations were verified post-experimentation. Given this, we are confident that the recording site in the current study is in the NCL. This is now clarified on line 185–187.

Line 247-250: because the authors do not know what exactly the crow had perceived, this sentence should be written as a suggestion and not as a statement.

We thank the reviewer for their suggestion. We agree that our interpretation should be stated as a suggestion rather than a definitive statement. We have revised the sentence to reflect this (lines 210–212).

Line 250-253: I don't understand the reason for why a behavioral response that occurred prior to the appropriate wait time was considered correct. The explanation given here sounds completely random.

We appreciate the reviewer's concern and would like to clarify our reasoning. Regardless of whether the crow received a reward (for waiting beyond the minimum time delay) or not, at the time of response, the subject had perceived that the appropriate amount of time had passed—otherwise, it would not have initiated a response. To ensure consistency across all three wait periods, we applied a uniform criterion that allowed responses occurring up to 20% prior to the

target time to be considered correct. This threshold was chosen because it captured an appropriate proportion of trials under the response time distribution while maintaining fairness and comparability across different wait durations.

Line 260-262: Here, the percentage of neurons considered as “increasing” and as “decreasing” should clearly be stated.

As noted in a previous response, in response to comments from another reviewer regarding this set of analyses, we conducted an alternative analysis focusing specifically on increasing neurons. As part of these revisions, we have now included the number of increasing neurons in the results section of the manuscript (**line 250, lines 475**).

Line 295-298: the numbers indicate that for 95 neurons a significant effect was found in both analysis windows. If this is the case, it should be mentioned.

In response to comments from another reviewer, we have updated our inclusion criterion, which has led to a slight adjustment in the total number of neurons included in our analyses. We have revised the manuscript and figures accordingly and have now explicitly reported the number of neurons that showed significant effects during both the cue offset and response onset periods (**lines 212–219, methods; lines 431–441, results**).

Line 300-301: what about neurons that showed a significant effect in both analysis windows? Was the preferred wait duration equal for both analysis windows? If not, which duration was chosen as the preferred duration? How many neurons showed a difference in preferred wait time between the two analysis periods? How was the number of neurons for each preferred wait time distributed over the two crows? Were neurons preferring short duration and long durations, respectively, equally frequently found in both animals?

We thank the reviewer for their thoughtful questions. In our analysis, we identified 75 neurons that showed significant effects during both periods of interest. Among these, 25 neurons exhibited a change in their preferred wait duration between the two periods. This information has been added to the revised manuscript (**lines 431–441**). Additionally, we have included a supplementary figure that shows the distribution of neurons preferring each wait duration, presented separately for each crow. This provides a clearer view of whether neurons preferring short or long durations were equally represented across the two animals (**new Supplementary Figure 1**).

Line 312-314: in how many neurons the activity for the preferred wait duration was significantly different from the activity during the two non-preferred wait durations?

It is important to note that a neuron is termed selective if at least one wait period elicited a significantly different firing rate compared to any other (ANOVA). There is no requirement that both non-preferred wait durations elicited significantly different firing rates. This approach is consistent with our previous studies in which we measured tuning curves to a variety of parameters, such as numbers. This approach is also consistent for all tuning curve assessments across sensory, cognitive, and motor domains.

Line 387: why was the analysis performed only on a subset of neurons? The criteria to exclude data needs to be mentioned.

We believe this comment is the same as the following and we answer it below.

Line 359: why was the analysis performed only on a subset of neurons? The criteria to exclude data needs to be mentioned.

As noted in lines 218–221 of the original manuscript, the state-space analysis was conducted on a subset of neurons because we applied a criterion requiring at least 30 trials per target duration for each neuron to be included. This detail now outlined on **line 388**.

Line 459-460: I strongly disagree with that statement. Without any other timing-relevant evidence, such as ramping of activity with elapsed time, the authors can't just conclude that the variation of neural activity with different waiting periods is indicating that NCL neurons encode time. There are many alternative explanations for NCL neurons to vary their activity according to instructed wait time.

We respectfully disagree. We believe our findings provide strong evidence that NCL neurons are functionally significant for timing behavior. Specifically, our results show that neural activity during the first 1200 ms of 1500 and 3000 ms trials reliably predicts whether the crow will wait the correct amount of time or make an error by waiting too long. This predictive relationship between early neural activity and subsequent behavior underscores the role of NCL neurons in time-related decision processes. Ramping of activity with elapsed time is certainly an important time code in many brain areas, but it is by no means the only possibility. We designed a controlled behavioural protocol to assess time estimation in the crows, and we would not be able to identify alternative explanations for why the neurons vary their activity according to the instructed wait time. However, to address the reviewer's concern, we have slightly reworded the sentence in the manuscript (**lines 724–726**) to clarify that the neurons encode the target duration rather than implying direct encoding of elapsed time. We believe this adjustment better reflects the data and maintains the core conclusion that NCL activity is critical for timing performance.

Line 591-592: why was it important that the wait and the go period were of the same duration?

We thank the reviewer for their question. We have clarified in **the Figure 1 caption** that the Wait and Go periods were matched in duration to manage the increase in response time variability associated with longer estimates. This consistency was important to ensure the accuracy of timing estimates across trials. Additionally, in the caption, we now specify that the figure displays a 1.5-second trial example for clarity (**Figure 1**).

Figure 2: it should be specified what time the x-axis label is referring to. Time relative onset of go-period? The bin-size of the histograms should be indicated in the figure caption. The dotted vertical line indicating the cued time interval can hardly be seen in the plot.

We thank the reviewer for their suggestion. We have updated the **Figure 2 caption** to clarify that the x-axis represents the subject's time estimation, measured as time since cue offset. Additionally, we have adjusted the figure to make the dotted vertical line indicating the cued time interval more visible. We hope these changes improve the clarity of Figure 2.

Figure 3: without any indication of the locations of recording sites, panel A is not very informative. Panel B and C should be swapped. According to the figure caption, panel B shows a neuron selective for 1.5 s, but in panel B, the red activity profiles demonstrate lowest firing rates. The grey shading is hardly visible in panels B-D. In the raster plot for the 6s wait time trials in panel D, the neuron seems to not spike at all during many subsequent trials, while there were some spikes in other trials. A complete lack of neural activity during some trials but not during others can be a sign of a systematic failure of the recording equipment. Was the lack of activity specific to the wait period or was this also seen during pre-cue and cue periods of the specific trials?

As mentioned in response to a previous comment, in alignment with the 3Rs principle to reduce the number of animals used in research, we use the same crows across multiple experiments and do not sacrifice them after each study (**line 185–187**). While post-experiment sacrifice and histological processing would enable precise confirmation of neural recording locations, exact locations were not obtained for this specific experiment. We included this panel to provide an overview of the location of the NCL within the avian brain, as we feel it offers anatomical context for readers.

We appreciate the reviewer's observation regarding the misalignment between the figure and caption for Panels B and D. We have corrected this by **updating the Figure 3 caption** to match the correct panel order. Additionally we have darkened the grey shading in the raster plots, which we hope improves the figure's readability.

As for the pattern of neural activity in Panel D, the trial types shown are grouped for visualisation purposes and do not reflect the actual sequence in which they occurred. If there had been a fault in the recording equipment, we would expect to observe similar instances of no activity across different trial types (e.g., red or blue trials). However, the neuron's activity remained consistent across trials, even though longer trials are less frequent.

Figure 4: The authors may want to change the color pattern in this figure, to enable color-blind people access to the data. There is a typo in the Y-axis label in panel A and C.

We appreciate the reviewer's suggestion regarding colour accessibility. The current colour scheme was chosen to reflect the actual colours used in the experiment, as these are directly related to the stimuli the birds encountered. We feel it is important to maintain this alignment between the figure and the experimental conditions to accurately convey the context of the study. However, we will ensure that the figure includes clear labels and patterns to help make the data accessible to all readers, including those with colour vision deficiencies.

We thank the reviewer for pointing out the typo, and we have corrected it in the revised figures.

2. Revision of Nature Communications manuscript NCOMMS-24-50946A

Reviewer #1

The authors answered properly all the reviewers' and now the paper delivers a strong take home message.

Minor comments:

Please refer to the paper by Manuel Beiran et al., 2023 untitled: Parametric control of flexible timing through low-dimensional neural manifolds. This paper is important for the present study because using recurrent neural networks, they found that confining the dynamics to a low-dimensional subspace allowed tonic inputs to parametrically control the overall input-output transform, enabling generalization to novel inputs and adaptation to changing conditions. The tonic inputs they used are similar to your tonic interval tuned neurons.

We thank the reviewer for this helpful suggestion. In the revised Discussion section, we now refer to the work by Beiran et al. (2023) and explicitly draw the conceptual link between their tonic control inputs and the tonic interval-tuned neurons identified in the NCL (lines 391–399).

Typo: Single-unit correlates of time estimation

We thank the reviewer for highlighting this typo. We have corrected it in the revised manuscript (line 166). Additionally, we have slightly adjusted the section title to better reflect the updated analyses and the overall restructuring of the manuscript.

Reviewer #2

The revised manuscript from Johnston, Kirschhock, and Nieder is highly responsive to the reviewer comments. New analyses were added to address many of the questions raised, and extensive changes were made to the text and figures. Overall, the paper is much stronger, but there are still a few remaining and new issues.

The authors added a new analysis based on an SVM classifier (lines 267-286 and 503-511) to test the alternative hypothesis that NCL neurons encode elapsed time (as opposed to intended waiting time, which is their main hypothesis). I think the analysis is basically sound, but the description in the Methods is quite confusing, and the motivation could be explained more. Because SVM classifiers are employed in different ways throughout the manuscript, one helpful clarification would be to explicitly state what the features and categories are for each analysis. The underlying question seems to be whether the average firing rate of a neuron varies enough over time and is consistent enough across trials for an SVM classifier to decode what proportion of the interval has elapsed. As the classifier makes no assumptions about how the rate varies, it's a more general analysis than the linear/exponential/sigmoidal fit or the time-rescaling analysis and so it would make more sense coming first in this section rather than last. Some important details seem to be missing: how does the classifier performance vary across the population, and how are the data shown in Figure 5 related to the output of the model? Are the median and interquartile ranges across neurons, bootstrap replicates, or validation folds?

We are grateful to the reviewer for these thoughtful and constructive comments. In the revised Methods section, we have now explicitly defined the features (neuronal firing rates in each time bin) and category labels (bin numbers corresponding to elapsed time) used in the SVM classifier (**lines 612–643**). Additionally, we clarified that although classifiers were trained individually on each neuron's data, decoding performance was aggregated across neurons, making this fundamentally a (sub-)population-level decoding analysis.

To further address the reviewer's point about the population perspective, we repeated the SVM decoding analysis using the full recorded population and included these results as **new Supplementary Figure 3**. This supplementary analysis confirms that elapsed time can be decoded robustly even when pooling across all recorded neurons.

We acknowledge that the SVM analysis was at times incorrectly referred to as a "single-cell" analysis in the original manuscript. We apologize for any confusion this may have caused and have now carefully revised the manuscript to consistently describe this work as a (sub-)population-level decoding analysis. We emphasize that this was an honest oversight rather than an attempt to mislead, and we hope the corrections now clarify both the structure and interpretation of the analysis.

We have also clarified how the data in Figure 5 relate to the model outputs. Specifically, the trained SVM models predicted elapsed time bins for all test observations, with each true bin represented four times per fold. Predicted labels were accumulated across all cross-validation folds and 1000 repetitions, resulting in 20,000 predicted values per true bin. These distributions were then summarized by the median and interquartile range across predicted elapsed times for each bin, and these summaries are visualized in Figure 5.

Regarding the organization of the Results section, we would like to clarify that while the overall order of results remains the same, we have now positioned the SVM analysis of elapsed time decoding as the first of the population-level analyses. This adjustment highlights the shift from single-neuron findings to population-level decoding, while maintaining the original logical flow of the manuscript.

The authors adequately addressed my concern about selection bias in their rebuttal and by showing more data for individual neurons (Figure 3E,F is very helpful here). For tuning curve width, I am not sure that fitting a Gaussian function to three data points is the best approach, because there may be a lot of error and numerical instability in the nonlinear fitting procedure. Calculating the sample standard deviation across the three points should be more numerically stable.

We appreciate the reviewer highlighting this important point regarding the numerical stability of the tuning curve fitting approach. In response, we revised our analysis of tuning curve width. We now calculate the weighted standard deviation of the x-values for each tuning curve, using the corresponding normalized response values (y-values) as weights (**lines 576–579** and **150–152** for methods and results, respectively). This provides a more numerically stable and interpretable measure of tuning width without relying on fitting procedures. We also **updated Figure 4C** and **4F** to reflect the results of this revised analysis. Additionally, we used a Jonckheere-Terpstra trend test ($\alpha = 0.05$) to assess whether tuning curve widths increased systematically across target durations.

The analysis of error trials is interesting and important but also somewhat difficult to follow in the methods and results. Why were the “late-error” but not the “early-error” trials analyzed? Looking at the response time histograms, it seems like there should be quite a few early-error trials for each category but almost no late-error trials (e.g., there are no blue bars after 6000 ms). Are these trials where the bird doesn’t respond at all? In that case, the failure of the classifier could reflect the birds not being engaged in the task rather than an error prospectively encoding the interval.

We thank the reviewer for highlighting the need for clarification. We have now revised the Methods and Results sections to better explain our rationale for focusing on late errors only (lines 667–674 and 244–246) for methods and results, respectively). Early errors were excluded because their causes are heterogeneous—they may reflect either genuine timing misjudgments or intentional trial abortion, making them difficult to interpret consistently. In contrast, late errors were clearly defined as trials in which the animal did not respond within the allotted response window, and these trials were only included when there was clear evidence of continued task engagement (i.e., the bird maintained the required posture throughout the trial). This ensures that late errors reflect genuine failures in time estimation.

Minor issues:

I. 31: “the executive telencephalon of male crows” implies that female crows might have a different executive telencephalon, which is probably not intended; please reword.

To avoid the unintended implication, we have revised the manuscript by moving the mention of “male” earlier in the paragraph, where we describe the subjects. This makes it clear that both animals used were male, without suggesting a sex-specific difference in telencephalon structure (line 28).

I. 205: Criteria used to determine if clusters were single units (e.g. signal-to-noise ratio, some index of cluster separation from noise, proportion of refractory period violations in the autocorrelations) is still missing.

In the revised Methods section, we have now included additional details regarding the criteria used to define single units (lines 541–546).

I. 214: “trials” instead of “responses” would be clearer and more consistent with rest of the paper.

We agree and have replaced “responses” with “trials” to improve clarity and maintain consistency throughout the manuscript (line 555).

I. 412: Why was the mean of the Gaussian manually specified instead of being fit from the data?

We fixed the center of the Gaussian fit to the peak of the response time histogram to ensure stable and representative fits, particularly in cases where the distribution was skewed by early responses (e.g., 6000 ms trials). Allowing the mean to vary sometimes led to poor fits that did not reflect the behaviorally relevant central tendency. Since the goal of this analysis was to characterize the spread of response times rather than to precisely estimate the mean, anchoring the fit to the mode provided a more robust basis for comparison across target durations. We have updated the manuscript to clarify this point (lines 97–101).

I. 461: Recommend using past tense throughout Results.

Indeed, we have revised the Results section to ensure consistent use of the past tense throughout. All instances of present or present perfect tense have been corrected accordingly.

I. 465: monotonicity -> monotonic

I. 466, 468: “monotonous” means boring. I think you mean “monotonic”

We thank the reviewer for catching this. We have corrected the term “monotonous” to “monotonic” in the revised manuscript (lines 154, 156).

I. 485: “scalability of neuronal activity” could mean rescaling in time or in amplitude, please be specific.

We agree that this requires clarification. We have revised the text to specify that we refer to temporal scaling of neuronal activity, and not amplitude rescaling (line 173).

Figure 3E,F: the plots show about 260 neurons, which doesn't correspond to the total number of units (409) or the number of time neurons (238). Please specify what neurons are shown. Also, what is 0 in the normalization? The minimum rate for each neuron, or 0 Hz?

We thank the reviewer for pointing out this discrepancy in Figure 3E,F. There was an error in the y-axis labeling, which originally showed "240" as the top value—it should have been "220". This has now been corrected in the **updated figure 3E,F**. The figure includes only the 220 “time” neurons that showed a significant effect of time in either the cue-offset or response-onset alignment period. Neuronal activity was normalized within each neuron, such that 0 corresponds to the minimum firing rate observed for that neuron across the time window, and 1 corresponds to the maximum firing rate. We have revised the figure caption to make both points explicit (lines 937–940).

I. 724: arcopallium, not archipallium

We appreciate the reviewer's attention in identifying this terminology error. We have corrected “archipallium” to “arcopallium” in the revised manuscript (line 417).

I. 732: with apologies for not catching this earlier, I don't think it's accurate to say that the avian pallium doesn't have layers, it's just that they're not as distinct, universal, or formed in the same way as in the mammalian isocortex.

We have revised the text to reflect that the avian pallium is not unlayered, but rather exhibits a different organizational structure compared to the mammalian isocortex (lines 424–425).

Reviewer #3

I thank the authors for thoroughly revising their manuscript. Now, it reads like a very interesting and convincing story. Especially the comparison of neural activity between correct and incorrect trials and the new result on a possible sequential representation of time by ensembles of NCL neurons are exciting. All my comments were fully addressed and the manuscript can now be accepted for publication.

Line-specific comments:

Line 31: the word “male” should be moved to line 27 (We trained two male crows ...) otherwise it reads as if only male crows possess an executive telencephalon.

We thank the reviewer for this suggestion. We have moved the reference to “male” earlier in the paragraph to clarify that both experimental subjects were male, without implying that only male crows possess an executive telencephalon (**line 28**).

Line 184: please specify how many microdrives were implanted per bird and per hemisphere, and how many electrodes contained each microdrive?

We have now clarified in the manuscript that each implant contained two microdrives, and that each microdrive held four electrodes. We also specify the hemisphere. These changes have been added to the Methods section (**lines 515–520**).

Line 191: By what distance was the electrode moved on average each day? Were all electrodes in one drive moved simultaneously or independently?

We appreciate the reviewer’s comment and have now included additional methodological details in the revised text. Specifically, we clarified that each implant contained two microdrives, with each microdrive advancing four electrodes simultaneously. Microdrives were adjusted independently depending on signal quality. Based on the drive’s mechanical specifications in Crow 1, the microdrives were moved on average 93.6 $\mu\text{m}/\text{day}$ and 67.4 $\mu\text{m}/\text{day}$, and in Crow 2, 91.7 $\mu\text{m}/\text{day}$ and 105.6 $\mu\text{m}/\text{day}$ (**lines 528–531**).

Line 558 – 560: Could you please show an example of a response of one neuron to a specific target time during a correct trial and during an incorrect trial? I would be interested in seeing whether there are already differences in activity at the single neuron level.

We appreciate the reviewer’s interest in observing single-neuron responses across correct and incorrect trials. Unfortunately, due to the low number of late-error trials per neuron, we were unable to identify a representative single unit with enough error trial repetitions to meaningfully illustrate activity differences at that level. For this reason, and because the analysis was designed to assess population-level differences in encoding between correct and late-error trials, we have focused on aggregated decoding performance across neurons. We believe this approach offers a more robust and interpretable measure of how time-related activity differs depending on behavioral outcome.

Line 644: I suggest to rephrase this headline to make it not sound so negative. Maybe: “Sequential encoding of time by NCL neuron ensembles”

We thank the reviewer for the helpful rephrasing suggestion. We have updated the section heading to “Sequential encoding of time by NCL neuron ensembles” as recommended (**line 335**).

Figure 4 A and D are not mentioned in the text

We thank the reviewer for pointing this out. We had incorrectly labeled the figure references in the original text. This has now been corrected, and **Figure 4A** and **4D** are explicitly referenced in the revised manuscript (**lines 137 and 139**).

Revision of Nature Communications manuscript NCOMMS-24-50946B
Reply to reviewers

Reviewer #2 (Remarks to the Author):

The authors have adequately addressed all of my concerns, and the paper is ready for publication. This is a strong contribution to the field, and the methodology is sound.

Thank you

Reviewer #3 (Remarks to the Author):

I thank the authors for addressing all my remaining comments and congratulate them on a great paper.

Thank you